# Condensin controls recruitment of RNA polymerase II to achieve nematode X-chromosome dosage compensation

**William S Kruesi[1†], Leighton J Core[2†], Colin T Waters[2], John T Lis[2], Barbara J Meyer[1]\***

[1]Department of Molecular and Cell Biology, Howard Hughes Medical Institute, University of California, Berkeley, Berkeley, United States; [2]Department of Molecular Biology and Genetics, Cornell University, Ithaca, United States

**Abstract** The X-chromosome gene regulatory process called dosage compensation ensures that males (1X) and females (2X) express equal levels of X-chromosome transcripts. The mechanism in *Caenorhabditis elegans* has been elusive due to improperly annotated transcription start sites (TSSs). Here we define TSSs and the distribution of transcriptionally engaged RNA polymerase II (Pol II) genome-wide in wild-type and dosage-compensation-defective animals to dissect this regulatory mechanism. Our TSS-mapping strategy integrates GRO-seq, which tracks nascent transcription, with a new derivative of this method, called GRO-cap, which recovers nascent RNAs with 5′ caps prior to their removal by co-transcriptional processing. Our analyses reveal that promoter-proximal pausing is rare, unlike in other metazoans, and promoters are unexpectedly far upstream from the 5′ ends of mature mRNAs. We find that *C. elegans* equalizes X-chromosome expression between the sexes, to a level equivalent to autosomes, by reducing Pol II recruitment to promoters of hermaphrodite X-linked genes using a chromosome-restructuring condensin complex.

**\*For correspondence:** bjmeyer@berkeley.edu

†These authors contributed equally to this work

**Competing interests:** The authors declare that no competing interests exist.

**Reviewing editor**: Nick Proudfoot, University of Oxford, United Kingdom

## Introduction

The essential, X-chromosome-wide regulatory process called dosage compensation ensures that males (XO or XY) and females (XX), from worms to humans, express equivalent levels of X-chromosome products despite their unequal dose of X chromosomes (*Gelbart and Kuroda, 2009*; *Meyer, 2010*; *Conrad and Akhtar, 2012*; *Jeon et al., 2012*). The failure to dosage compensate is lethal. Dosage compensation strategies differ across species, but invariably a regulatory complex is targeted to the X chromosomes of one sex to modulate transcription along the entire X. The molecular mechanisms by which these complexes regulate gene expression remain elusive. Here we analyzed X-chromosome dosage compensation in the nematode *Caenorhabditis elegans* to determine the step of transcription controlled by its dosage compensation complex (DCC). The DCC binds to both X chromosomes of hermaphrodites to reduce transcription by half (*Meyer, 2010*; *Pferdehirt et al., 2011*). Sequence-specific DNA binding sites recruit the DCC to X and facilitate its spreading along X (*Ercan et al., 2009*; *Jans et al., 2009*; *Pferdehirt et al., 2011*). The DCC shares subunits with condensin (*Csankovszki et al., 2009*; *Mets and Meyer, 2009*), a protein complex required for the compaction, resolution, and segregation of mitotic and meiotic chromosomes (*Wood et al., 2010*), suggesting that DCC-dependent changes in chromosome structure facilitate transcription regulation. In principle, the DCC could control any step of transcription: recruitment of RNA polymerase II (Pol II) to the gene promoter, initiation of transcription, escape of Pol II from the promoter or proximal pause sites, elongation of RNA transcripts, or termination of transcription.

**eLife digest** In many species, including humans, females have two X chromosomes whereas males have only one. To ensure that females do not end up with a double dose of the proteins encoded by genes on the X chromosome, animals employ a strategy called dosage compensation to control the expression of X-linked genes.

The mechanisms underlying dosage compensation vary between species, but they typically involve a regulatory complex that binds to the X chromosomes of one sex to modify gene expression. In the nematode worm *Caenorhabditis elegans*—which consists of hermaphrodites (XX) and males (XO)—this regulatory complex, called the dosage compensation complex (DCC), binds to both X chromosomes of XX individuals, reducing gene expression from each by 50%. DCC shares many subunits with a protein complex called condensin, which regulates the structure of chromosomes to achieve proper chromosome segregation. However, it is unclear exactly how the DCC controls the expression of X-linked genes.

For a gene to be expressed, an enzyme called RNA polymerase II must bind to the gene's promoter—a stretch of DNA upstream of the protein-coding part of the gene—so that it can begin transcribing the DNA into RNA. Promoters have been difficult to define in *C. elegans*, but Kruesi et al. devised a strategy to map transcription start sites, and hence promoters, throughout the worm genome. The strategy integrates the results of two methods: One measures the extent and orientation of each gene's transcribed region, and the other locates the distinctive cap structures that mark the true 5' ends of newly made RNAs.

Using this new promoter information, coupled with genome-wide measurements of the levels of newly synthesized transcripts from wild-type and dosage-compensation-defective animals, they showed that *C. elegans* achieves dosage compensation by reducing the recruitment of RNA polymerase II to the promoters of X-linked genes in XX individuals.

Kruesi et al. also identified a second regulatory mechanism that acts in both sexes to increase the level of transcription of genes on the X chromosome. This ensures that after dosage compensation, genes on the X chromosome are expressed at a similar level to those on the autosomes (all chromosomes other than X and Y).

As well as shedding light on the mechanism by which dosage compensation occurs in *C. elegans*, the study by Kruesi et al. provides a valuable data set on transcription start sites in the worm, and puts forward a general strategy that could be used to map these sites in other species.

To understand the mechanism of *C. elegans* dosage compensation, we first developed a procedure to map the position, density, and orientation of transcriptionally engaged Pol II genome-wide in *C. elegans* and then devised a strategy to identify the transcription start sites (TSSs). Nascent RNA transcripts from approximately 70% of *C. elegans* genes undergo a rapid co-transcriptional processing event in which the 5' end is replaced by a common 22-nucleotide leader RNA (SL1) through a trans-splicing mechanism (*Blumenthal, 2012*). Because trans-splicing removes information about Pol II initiation from nascent RNAs, TSSs have been difficult to identify from accumulated mRNAs (*Morton and Blumenthal, 2011*). The paucity of annotated promoters has made transcription regulation a challenge to study in *C. elegans*. By comparing the quantity, location, and direction of engaged Pol II from wild-type and dosage-compensation-defective embryos relative to TSSs, we determined the step of transcription controlled by the DCC.

Our work establishes a general strategy for TSS mapping in any organism and provides an invaluable TSS data set for dissecting *C. elegans* gene regulation. We show that *C. elegans* equalizes X-chromosome-wide gene expression between the sexes by reducing Pol II recruitment to the promoters of X-linked genes in XX embryos via a mechanism that utilizes a chromosome-restructuring complex. We also show that a separate regulatory mechanism functions in *C. elegans* to elevate the intrinsic level of transcription from the X chromosomes of both sexes, so that after dosage compensation, X chromosomes and the two sets of autosomes have equivalent expression.

## Results

### Genome-wide mapping of transcriptionally engaged Pol II and transcription start sites reveals promoters to be far upstream of mature mRNA 5′ ends

To map the distribution of transcriptionally engaged Pol II genome-wide, we performed global run-on sequencing (GRO-seq) experiments using nuclei from three stages of wild-type animals (embryos, starved L1 larvae, and L3 larvae) and dosage-compensation-defective embryos. In GRO-seq reactions, engaged polymerases are allowed to transcribe (run-on) short distances (100 nucleotides) and incorporate affinity tags into their nascent RNAs under conditions that prohibit new initiation (*Core et al., 2008*). Tagged transcripts are affinity purified, amplified, sequenced, and aligned to the genome to map engaged Pol II (*Figure 1A–E*).

The two GRO-seq biological replicates for each stage had high statistical correlation throughout the genome (Spearman correlation, ρ > 0.94) and across gene bodies (Spearman correlation, ρ > 0.98) (*Figure 1—figure supplements 1–3* and *Figure 1—source data 1*). Gene expression levels calculated from GRO-seq data correlated well with expression data from microarrays and RNA-seq experiments (*Figure 1—figure supplement 4*). For the majority of expressed genes, we found continuous GRO-seq signal upstream of the WormBase (WB)-annotated transcription starts (*Figure 1A,B*), suggesting that GRO-seq reactions contain nascent RNAs with true 5′ ends.

To map TSSs unambiguously, we performed a series of enzymatic selections on our GRO-seq run-on RNAs to capture only those RNAs with a 5′ cap, the 7-methyl guanosine residue added shortly after transcription initiation (*Rasmussen and Lis, 1993*). This modified GRO-seq procedure, called GRO-cap (*Figure 2*), enabled us to map TSSs with nucleotide resolution by tracking nascent RNAs prior to trans-splicing. Use of nascent RNAs without size selection also reduces the background from RNAs that are capped post-transcriptionally and increases the probability of identifying TSSs from promoters with low Pol II occupancy.

For genes that lack trans-splicing, the site of maximum GRO-cap signal was coincident with both the 5′-most GRO-seq signal and the WB-annotated start (*Figure 1C*), confirming that GRO-cap and GRO-seq together permit high-confidence mapping of TSSs. For trans-spliced genes with no previously identified TSSs, strong GRO-cap and GRO-seq signals were found upstream of the WB-annotated starts, and TSS calls were supported by uninterrupted GRO-seq signal between the maximum GRO-cap signal and the WB start (*Figure 1A*).

In total, a TSS was identified for 31.7% (6353 genes) of all *C. elegans* protein-coding genes from at least one of the three developmental stages examined (*Figure 1—source data 2*). Of our TSS calls, 77% are for genes shown previously to be trans-spliced (*Figure 1—source data 2*) (*Allen et al., 2011*). Plotting the average GRO-seq and GRO-cap signals from each developmental stage relative to the TSSs from the same stage revealed the vast improvement in gene models (*Figure 1F,G* and *Figure 1—figure supplements 5, 6 and 7A–C*). Independent GRO-cap reactions from the same developmental stage gave very similar results (*Figure 1G* and *Figure 1—figure supplement 7D*).

TSSs were also annotated for the majority of genes encoding short non-coding RNAs such as snoRNAs, 21 U-RNAs, and microRNAs, including TSSs for the five polycistronic microRNA clusters (*Figure 1C,E* and *Figure 1—source data 3*) (see 'Materials and methods'). The TSSs for the 21 U-RNAs were 2 bp upstream from the mature RNA (*Figure 1—figure supplement 9*).

In addition, the primary TSSs and gene composition were determined for many multigenic transcription units called operons in which the polycistronic pre-mRNAs are processed to monocistronic mRNAs through 3′ end formation and trans-splicing using SL2 RNA (*Figure 1D,E*) (*Blumenthal, 2012*). Identification of the TSS for one operon showed it to include a 5′ gene not previously ascribed to it (*Figure 1D*). TSSs for several operons showed the TSS to be far downstream (>2 kb) of the WB start, a result we also found for genes not included in operons (*Figure 1—figure supplement 10A,B*). Within operons some genes have independent promoters to transcribe their pre-mRNAs (*Allen et al., 2011*), and their TSSs were determined, including snoRNA genes within introns of internal genes (*Figure 1E* and *Figure 1—figure supplement 10A,C*). In general, we used conservative statistical criteria for calling TSSs, and additional TSSs can be identified by visual inspection of the data.

The new TSS calls revealed promoters to be further upstream from the 5′ ends of mature mRNAs than previously thought, as demonstrated by heat maps showing the GRO-seq signal or GRO-cap signal of re-annotated genes relative to WB starts or TSSs (*Figure 1F,G* and *Figure 1—figure supplement 7A–D*)

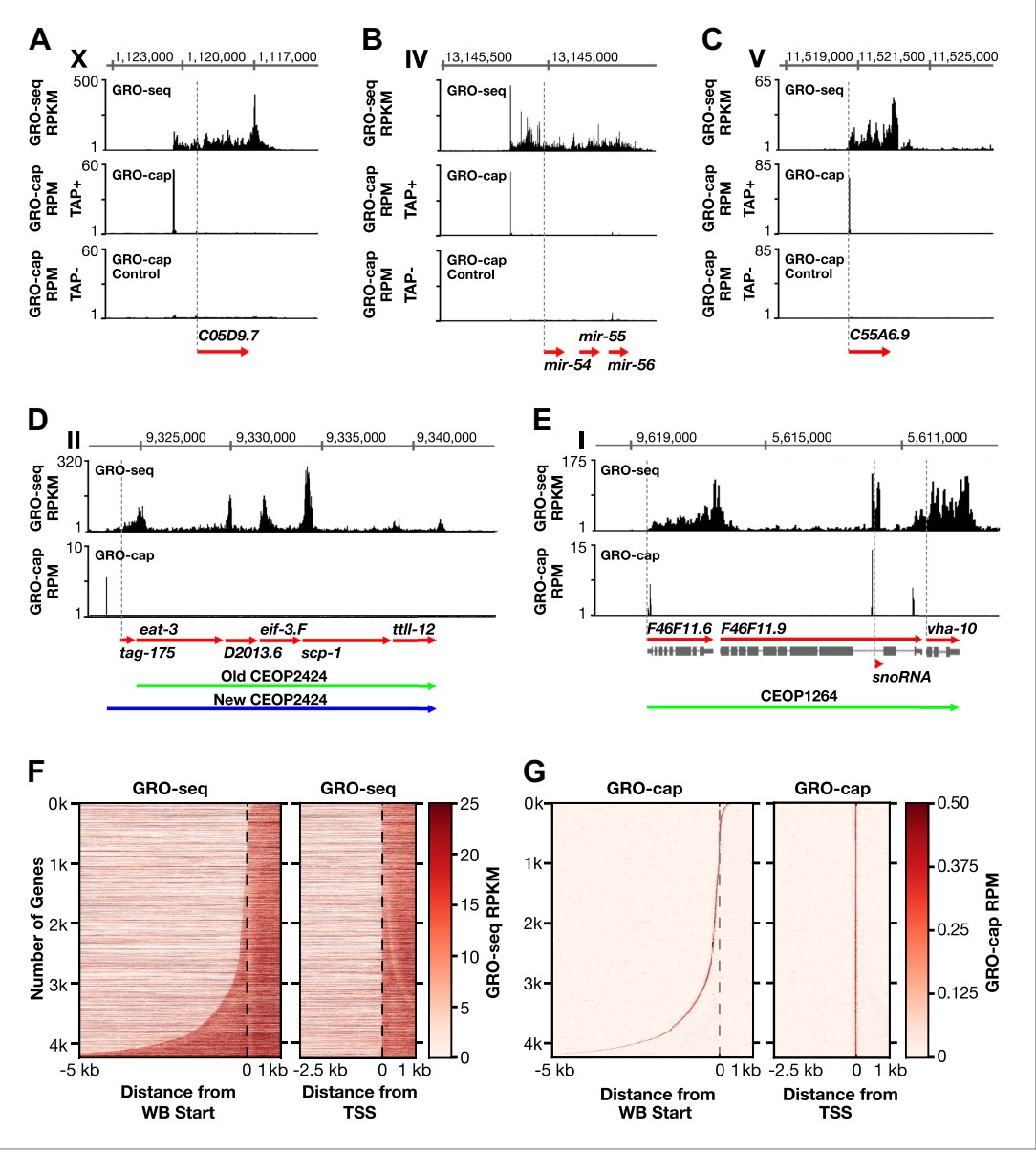

**Figure 1**. Genome-wide annotation of *Caenorhabditis elegans* transcription start sites. (**A**)–(**E**) Examples of newly annotated transcription start sites (TSSs) for protein-coding genes, non-coding RNA genes, and multigenic transcription units called operons identified using the combination of GRO-seq and GRO-cap. Red arrows demark the WormBase (WB) gene models. Dashed vertical lines show the WB gene starts. The GRO-seq signal is in reads per kilobase per million (RPKM). For protein coding genes, the GRO-seq signal was averaged across 25 bp windows with a 25 bp step. The GRO-cap signal is in reads per million (RPM). TAP+ is the signal from capped mRNAs, and TAP− is the background. For (**D**) and (**E**), the GRO-cap signal is the TAP+ signal after subtracting the TAP− signal. (**A**) TSS for a trans-spliced gene. The TSS maps 981 bp upstream of the WB start, with a continuous intervening GRO-seq signal. (**B**) TSS for the polycistronc microRNA cluster *mir-54-56* maps 158 bp upstream of the primary transcript start. (**C**) TSS for a non-trans-spliced gene. The TSS from GRO-cap and GRO-seq aligns with the WB start site. (**D**) Identification of the operon TSS shows the operon includes an additional gene, *tag-175*. The TSS for the operon maps 781 bp upstream of *tag-175*. (**E**) TSSs for genes in operons that also use independent promoters, including the TSS for a snoRNA gene within the intron of a gene. *vha-10* mRNA is trans-spliced with an SL2 RNA, indicating processing from a polycistronic RNA and an SL1 RNA, indicating transcription from an independent promoter. (**F**) Heat maps show that TSSs vastly improve gene models. The GRO-seq signal from embryos was plotted, one gene per row, for each of 4246 genes relative to the WB start (left) or the new TSS (right). The genes were ordered with increasing distance between the TSS and WB start. The light line moving rightward in
*Figure 1. Continued on next page*

*Figure 1. Continued*

the right panel does not represent TSSs. It reflects reduced GRO-seq signal immediately downstream of the trans-splice acceptor site that has been commonly annotated as the WB start site. (**G**) Heat maps showing the GRO-cap signal from embryos that was plotted for each of 4246 genes relative to the WB start (left) or the new TSS (right). The genes were ordered with increasing distance between the TSS and WB start.

The following source data and figure supplements are available for figure 1:

**Source data 1**. Aligned reads for GRO-seq, GRO-cap, and ChIP-seq experiments.

**Source data 2**. Annotation of transcription start sites for protein-coding genes.

**Source data 3**. Annotation of transcription start sites for non-coding RNAs.

**Figure supplement 1**. GRO-seq profiles are reproducible between replicates.

**Figure supplement 2**. Genome-wide GRO-seq signal is highly correlated between replicates.

**Figure supplement 3**. GRO-seq signal within protein coding genes is highly correlated between replicates.

**Figure supplement 4**. GRO-seq expression is correlated with gene expression from microarray and RNA-seq experiments.

**Figure supplement 5**. Genome-wide annotation of TSSs improves gene models.

**Figure supplement 6**. GRO-cap signal is strong at newly annotated TSSs.

**Figure supplement 7**. Heat maps showing GRO-seq and GRO-cap signal relative to either WB starts or TSSs for developmental stages reveal improvements in gene models.

**Figure supplement 8**. TSSs can be far upstream of the previously annotated WB starts.

**Figure supplement 9**. GRO-cap revealed that 21 U-RNAs have a TSS 2 bp upstream of the mature RNA.

**Figure supplement 10**. Features of promoters and TSSs.

**Figure supplement 11**. Distances between the TSS and WB starts of the trans-splicing acceptor site.

**Figure supplement 12**. Comparison of enhancers in *Chen et al. (2013)* and our annotated TSSs.

and histograms showing distances between TSSs and WB starts or SL1 trans-splice acceptor sites (TSA) (*Figure 1—figure supplement 11A,B*). The TSS-to-TSA, called the outron, was previously thought to be 50–500 bp. We found instead that outrons can be as long as 14 kb and have a median of 260 bp and mean of 753 bp (*Figure 1—figure supplement 11B*). Fully 59% of outrons are longer than 200 bp, 21% are longer than 1 kb, and 2.3% are longer than 5 kb (e.g., *Figure 1—figure supplement 8A–C*).

Multiple lines of evidence indicate that the GRO-seq signal between newly called TSSs and previously identified TSAs reflects legitimate outrons rather than independent overlapping upstream transcripts. First, 3′ UTRs or polyA signals are rare in outrons of >1 kb in length, indicating the engaged Pol II is not from independent polyadenylated transcripts. From 565 such outrons, only 1.4% had an identified 3′ UTR in the *Jan et al. (2011)* study, and 0.7% had a 3′ UTR in the *Mangone et al. (2010)* study. Furthermore, only 3.5% had a polyA site (*Mangone et al., 2010*). Second, regions corresponding to long outrons have a continuous ChIP-chip signal from antibodies enriched for either the ser2 phosphorylated form of Pol II or the hypo-phosphorylated form of Pol II (*Figure 1—figure supplement 8A–C*). These results and the restricted run-on length of ~100 nucleotides (*Core et al., 2008*) indicate that the GRO-seq signal corresponds to bound Pol II in vivo and is not an artifact of the nuclear run-on (NRO)

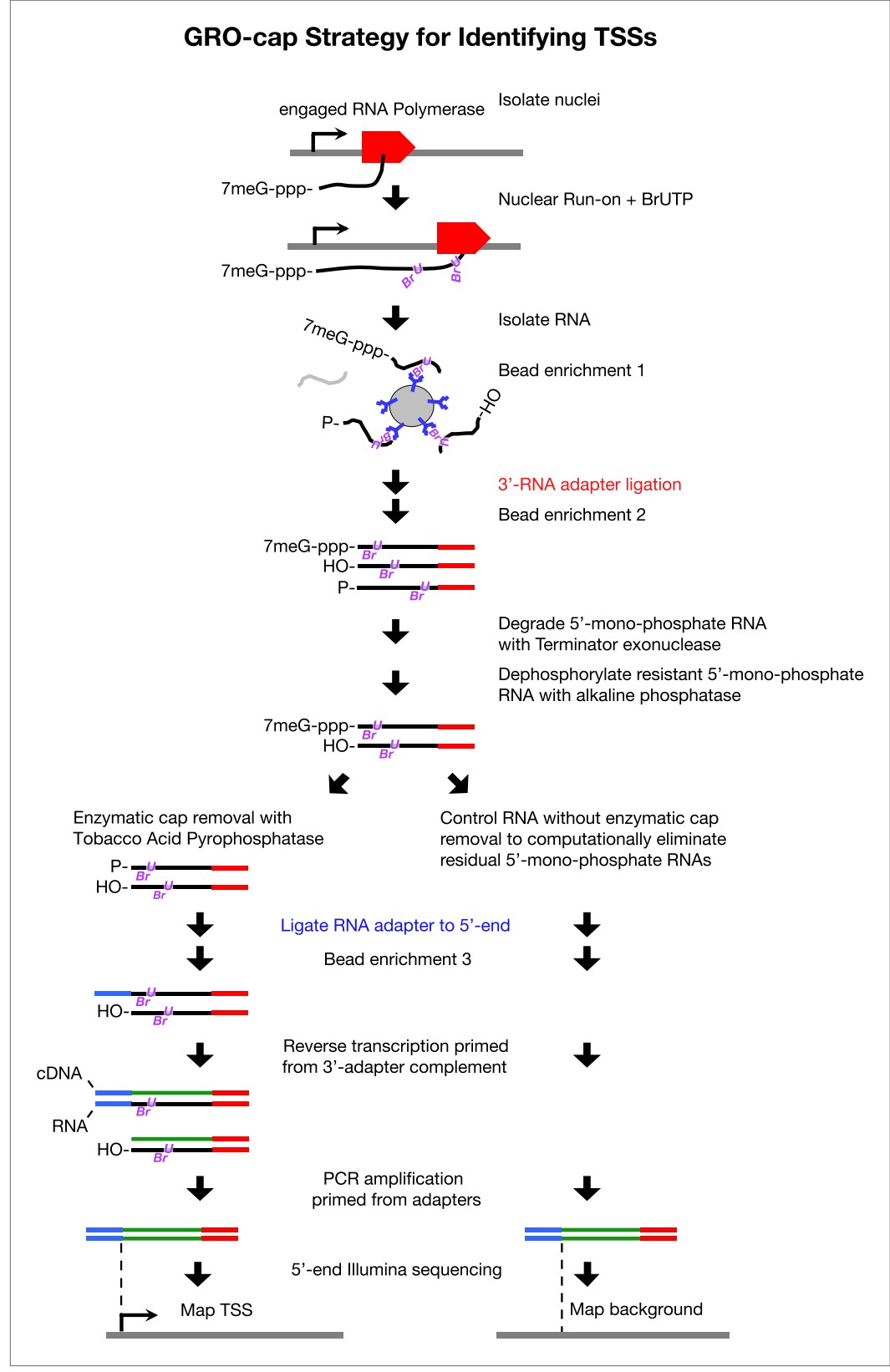

**Figure 2**. GRO-cap strategy for identifying TSSs. GRO-cap is a modified form of GRO-seq that utilizes the tagging and extensive purification of nascent RNAs from GRO-seq (***Core et al., 2008***) and then employs redundant enzymatic steps to enrich for RNAs with 5′ caps. Of particular importance to this study, GRO-cap permits analysis of *Figure 2. Continued on next page*

*Figure 2. Continued*

RNAs prior to their co-transcriptional processing, which replaces true transcription start sites (TSSs) with trans-spliced leader RNAs in *Caenorhabditis elegans*. GRO-seq run-ons have been tuned to extend the length of nascent RNAs by only 100 nucleotides on average, thus minimizing any possibility that independent transcription units might been artifactually linked. In GRO-cap, nuclei are isolated and RNA polymerases are allowed to transcribe briefly in a run-on reaction in the presence of Br-UTP, as in GRO-seq. RNA is isolated but the base-hydrolysis step of GRO-seq is omitted to increase the probability of capturing nascent RNA molecules with a 5′ 7-methyl-GTP cap. BrU-RNAs made during the run-on reaction are enriched by selection with anti-BrdU beads to ensure the identification of true TSSs from capped nascent RNAs rather than 5′ ends from RNAs that received post-transcriptional capping (*Fejes-Toth et al., 2009*). A 3′ RNA adapter (red) is ligated to the RNAs, followed by another round of bead enrichment. Selection against 5′ mono-phosphate RNAs that do not represent capped RNAs (and any carry-through 5′ RNA adapters) is achieved by sequential enzymatic treatment with Terminator exonuclease to degrade 5′ mono-phosphate RNAs and then alkaline phosphatase to remove 5′ phosphates from 5′ mono-phosphate RNAs resistant to the exonuclease. Half of the nuclear run-on (NRO) RNA pool is treated with tobacco acid pyrophosphatase (TAP+) to remove the 5′ cap from the RNA, thereby exposing a 5′ mono-phosphate. The other half is left untreated (TAP−) to provide a control population of residual 5′ mono-phosphate RNAs that never had 5′ caps. The 5′ mono-phosphate RNAs are ligated to 5′ RNA adapters (blue). The TAP+ and TAP− samples are prepared for Illumina sequencing as in GRO-seq by reverse transcription of RNA into DNA and then amplification of DNA from 5′ and 3′ adapter regions. We note that transcripts <500 bp are captured most efficiently on Illumina sequencing platforms. The enriched TSS regions are identified by mapping the 5′ ends of the sequence reads back to the genome and comparing the TAP+ and TAP− sites to eliminate false TSSs. Comparing the GRO-cap candidate TSSs to the 5′ ends of transcription units defined by GRO-seq permits reliable assignment of TSSs to transcription units.

reactions in vitro extending beyond the 3′ ends defined in vivo (*Figure 1—figure supplement 8A–C* and *Figure 3A*). Third, a heat map of individual genes showing the GRO-cap signal relative to TSSs reveals that a dominant TSS contributes the majority of the vast GRO-cap signal (*Figure 1G*). Together these observations strongly support the argument that the GRO-cap signal paired with the continuous GRO-seq signal from WB starts defines true TSSs. Recently published studies of TSSs in *C. elegans* that used cutoffs for TSS calls of either 1 kb upstream (*Gu et al., 2012*) or 200 bp upstream (*Chen et al., 2013*) of WB starts identified some outron TSSs but could not identify TSSs for a large class of genes with longer outrons (see 'Discussion').

## Features of *C. elegans* promoters

Several other noteworthy features of promoters emerged from this comprehensive mapping of *C. elegans* transcription units and TSSs. (1) Many trans-spliced genes have multiple TSSs (*Figure 3A*), suggesting that trans-splicing has removed the selective pressure to form promoters with a single TSS. (2) Genes can use different TSSs across developmental stages, indicating developmental stage-specific regulation of transcription initiation (*Figure 3B*). (3) DNA sequences flanking the newly annotated TSSs have strong sequence conservation across nematode species (*Figure 3—figure supplement 1B*) and also have evolutionarily conserved core promoter elements, including the TATA-box (worm consensus TATAWAWR) (*Figure 3C* and *Figure 3—figure supplements 1A and 2A,B*) and the initiator element (Inr) (worm consensus YCAYTY) (*Figure 3D* and *Figure 3—figure supplement 2C*), both of which facilitate formation of the Pol II pre-initiation complex (*Juven-Gershon and Kadonaga, 2010*).

## Features of *C. elegans* transcription: abundant 3′ Pol II accumulation and divergent transcription, but scarce promoter-proximal pausing

Three prominent features of transcription emerged. Accumulation of Pol II at 3′ ends of genes is abundant (*Figure 4A–C* and *Figure 4—figure supplement 1A,B*), as is divergent transcription from promoters lacking upstream divergent genes (*Figure 4D–F*). In contrast, promoter-proximal RNA Pol II pausing in *C. elegans* is rare under normal growth conditions, unlike in other metazoans, as shown later in 'Results'.

Pol II 3′ accumulation, likely caused by slow 3′ end formation and RNA processing (*Gromak et al., 2006*), is positively correlated in *C. elegans* with the expression level of the gene (*Figure 4—figure supplement 1A* and *Figure 4—source data 1*) and is more extensive in *C. elegans* than in *Drosophila* (*Figure 4A* and *Figure 4—figure supplement 1B*). Multiple peaks of 3′ accumulation within a gene

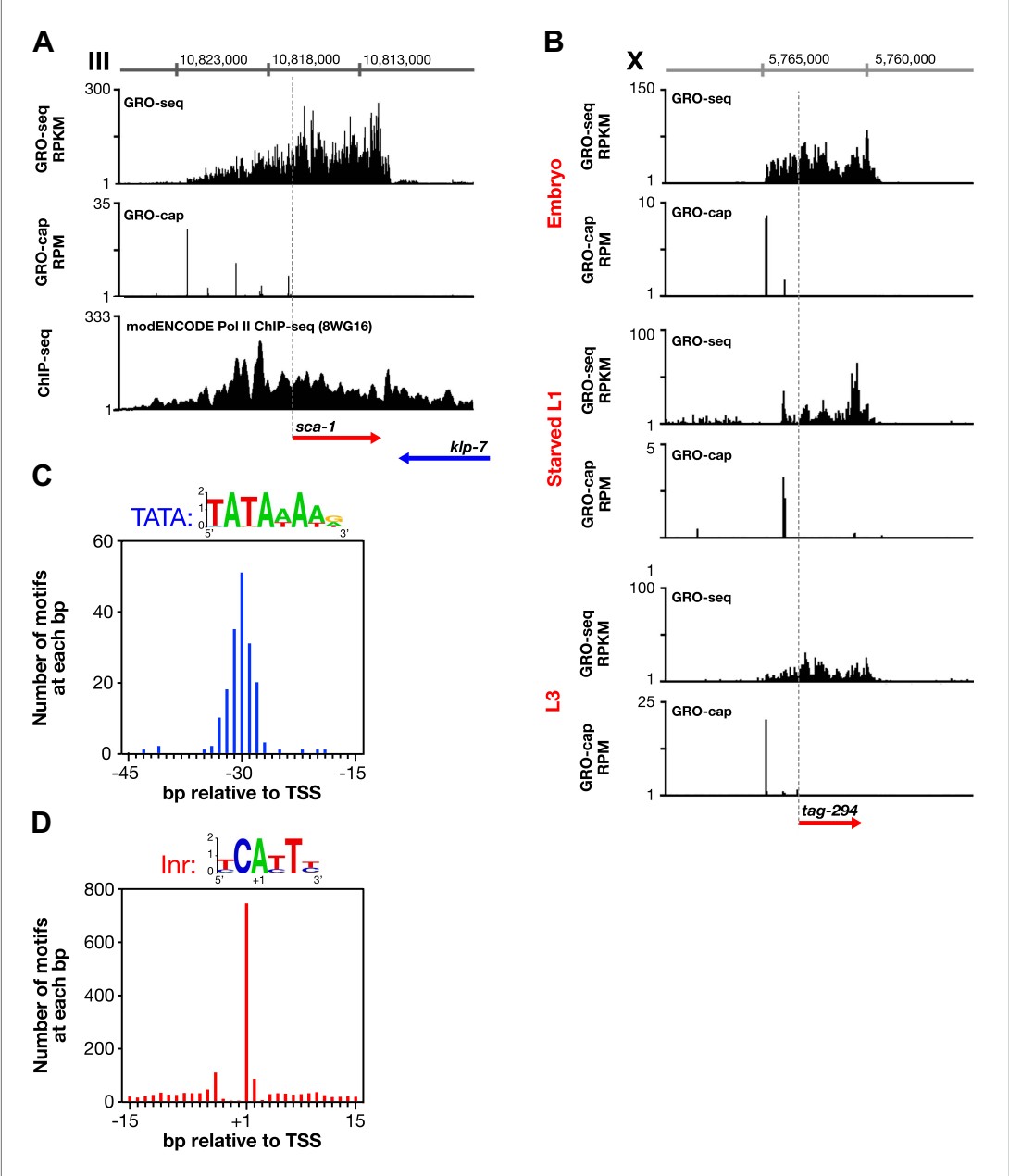

**Figure 3**. Features of promoters and TSSs. (**A**) Trans-spliced genes can have multiple transcription start sites (TSSs), suggesting that trans-splicing eliminates the pressure to have only one precise TSS per gene. Shown are GRO-seq and corrected GRO-cap (TAP+ signal after subtracting TAP− signal) signals with ChIP-seq data of hypo-phosphorylated Pol II (8WG16 antibody, modENCODE_2439) for the trans-spliced gene *sca-1* expressed in the L3 larval stage. The total GRO-seq signal becomes more intense as additional TSSs (from 5′ to 3′) contribute to the pool of engaged Pol II molecules that transcribe through the upstream regulatory region of *sca-1*. The combination of continuous Pol II signal in the upstream region and the lack of 3′ UTRs or polyA signals (**Mangone et al., 2010**) strengthens the interpretation that the GRO-cap signal combined with the continuous GRO-seq signal identified true TSSs for *sca-1*. From left to right, the TSSs reside upstream of the WormBase (WB) gene model by 5728 bp, 4582 bp, 3044 bp, 1669 bp, and 159 bp. The ChIP-seq signal 3′ of the *sca-1* 3′ UTR is from the *klp-7* gene on the opposite strand. (**B**) A gene can use different primary TSSs in different developmental states. The primary TSS for *tag-294* in embryos and L3 larvae is 1529 bp upstream of the WB start, while the primary TSS in starved L1 larvae is 656 bp upstream. DNA sequences flanking newly annotated TSSs have evolutionarily conserved core promoter elements, including (**C**) TATA-box elements and (**D**) initiator elements (Inr). Of 4547 embryo genes with TSSs, 162 genes (3.6%) have a TATA element with a perfect match to the consensus 15–45 bp upstream of it, and 745 genes (16.4%) have an Inr with the adenine residing at the TSS (+1 bp). Consensus sequences for TATA elements and the Inr are above the graphs. RPKM: reads per kilobase per million; RPM: reads per million.

*Figure 3. Continued on next page*

*Figure 3. Continued*

The following figure supplements are available for figure 3:

**Figure supplement 1**. Evolutionarily conserved promoter elements.

**Figure supplement 2**. Conserved core promoter elements in promoters of microRNA genes.

help identify genes having several isoforms with distinct 3′ ends (*Figure 1—figure supplement 10D*). The prevalence in *C. elegans* of polycistronic operons requiring extensive RNA processing to produce monocistronic mRNAs caused us to ask whether 3′ Pol II accumulation was positively correlated with the gene's position in an operon and hence the level of RNA processing. First and middle genes in an operon require two forms of co-transcriptional RNA processing at the 3′ end to generate monocistronic mRNAs, polyadenylation of the upstream gene, and trans-splicing of the downstream gene, while the terminal gene requires only polyadenylation at the 3′ end. First and middle genes showed more 3′ accumulation than terminal genes or genes not in operons (*Figure 4B,C*). In other words, 3′ pausing is longer at genes with another gene to process just downstream. The 3′ Pol II accumulation at terminal genes and genes not in operons was equivalent, yet greater than in *Drosophila* genes. Lastly, 3′ accumulation was similar between terminal genes of operons and monocistronic genes undergoing trans-splicing, and both gene sets had greater 3′ accumulation than monocistronic genes lacking trans-splicing, which nonetheless had more 3′ accumulation than *Drosophila* genes (*Figure 4—figure supplement 2A*). Cues triggering trans-splicing, particularly operon-specific trans-splicing, appear to facilitate 3′ accumulation and perhaps predispose *C. elegans* Pol II to greater 3′ pausing genome-wide, whether or not a gene resides in an operon. It should be noted that Pol II accumulation at 3′ ends does not overlap with U-rich regions at 3′ ends. Therefore, the high GRO-seq signal is not due to selective enrichment of U-rich RNAs (*Figure 4—figure supplement 2B,C*).

GRO-seq readily detects divergent transcription from promoters. In *C. elegans*, divergent transcripts are short and initiated 75–150 bp upstream from TSSs of promoters lacking upstream divergent genes (*Figure 4E* and *Figure 4—figure supplement 3A–C*). The frequency of upstream divergent transcription in *C. elegans* appears to be intermediate in degree between that in mammals, where it occurs at the majority of active promoters (*Kapranov et al., 2007*; *Core et al., 2008*; *Seila et al., 2008*), and that in *Drosophila*, where it occurs only rarely (*Nechaev et al., 2010*; *Core et al., 2012*) (*Figure 4F*). For both mammals and worms, promoters with TATA elements rarely support divergent transcription (*Figure 4—figure supplement 3D,E*) (*Core et al., 2012*). These results underscore fundamental similarities and differences in the architecture, evolution, and function of eukaryotic promoters.

## Genome-wide consequences of disrupting dosage compensation

Analysis of transcriptionally engaged Pol II in wild-type vs dosage-compensation-defective embryos provided a robust assessment of the genome-wide impact of disrupting dosage compensation. To assess dosage compensation, we disrupted *sdc-2* (sex determination and dosage compensation), the central XX-specific factor that triggers assembly of all DCC components onto X and induces hermaphrodite sexual differentiation by repressing the autosomal male sex-determining gene *her-1* (*Figure 5A*) (*Dawes et al., 1999*). Without *sdc-2*, DCC subunits fail to bind to X, and *her-1* is expressed, causing XX embryos to become severely masculinized and die from overexpression of X-chromosome genes (*Figure 5A*). The pivotal role of *sdc-2* in dosage compensation made its depletion the most effective way to disrupt dosage compensation, although depletion of any DCC condensin subunit causes similar XX lethality and elevation of X gene expression, as shown previously by our genome-wide measurements of gene expression (*Jans et al., 2009*).

Severe disruption of *sdc-2* function was achieved by treating an *sdc-2* partial loss-of-function XX mutant with *sdc-2* RNAi [*sdc-2(y93, RNAi)*] (*Figure 5—figure supplement 1*). GRO-seq data showed that expression of *her-1* (*Figure 5B*) and protein-coding genes on X (*Figure 5C*) were elevated in *sdc-2* mutants, while expression of genes on autosomes was mildly reduced (*Figure 5—figure supplement 2A–D*). *her-1* was de-repressed 12.7-fold, and the increase in GRO-seq signal was uniform from the

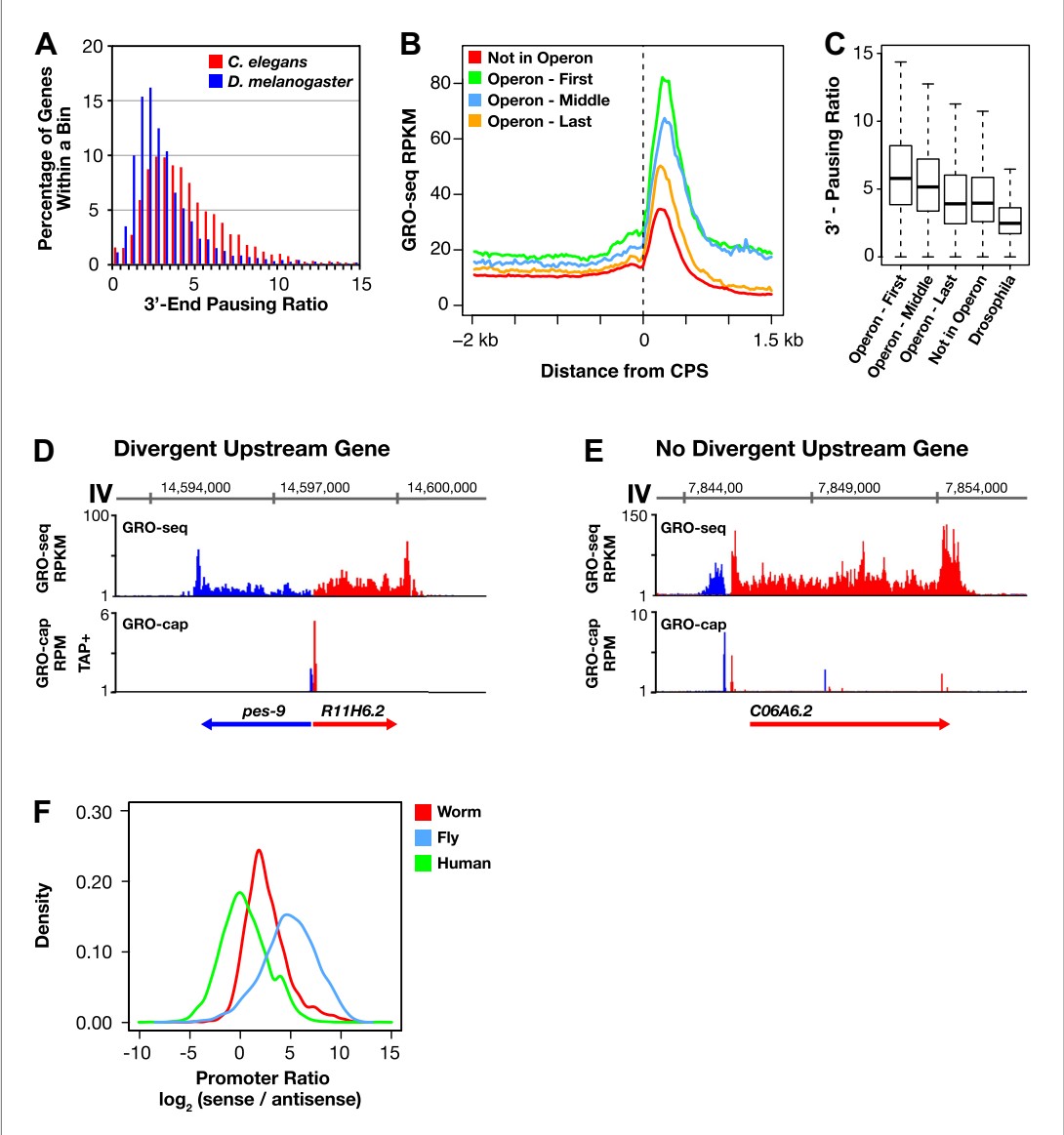

**Figure 4**. Features of *Caenorhabditis elegans* transcription: 3′ Pol II pausing and divergent transcription. (**A**)–(**C**) Pol II 3′ accumulation is prevalent in worms. 3′ End pausing ratios were calculated by dividing the highest average GRO-seq signal at the 3′ end by the average GRO-seq signal in the gene body. (**A**) A histogram of the 3′ end pausing ratios shows 3′ accumulation of Pol II is more extensive in *Caenorhabditis elegans* than in Drosophila. The histogram compares 3′ accumulation for 3984 genes expressed in *C. elegans* embryos with 6107 genes expressed in Drosophila cell lines. (**B**) The GRO-seq signal surrounding the 3′ end (cleavage and polyadenylation site [CPS]) was averaged for genes in or not in operons. Genes at the beginning (n = 430) and middle (n = 276) of operons were more highly expressed and had higher 3′ accumulation than genes at the end (n = 474) of operons or not in operons (n = 5048). Genes plotted had to be greater than 3 kb in length. (**C**) 3′ End pausing ratios were calculated for all classes of genes in (**B**) and plotted as boxplots. For this analysis, genes had to be greater than 3 kb in length and the gene body RPKM (reads per kilobase per million) had to be ≥1. Genes found at the beginning (n = 415) and middle (n = 275) of operons had higher 3′ pausing than genes at the end (n = 467) of operons. Genes lacking a downstream gene had similar 3′ pausing ratios whether or not (n = 3670) they were in operons. The 3′ pausing ratios for genes in all classes were greater than for Drosophila genes (n = 3260). (**D**)–(**F**) Upstream divergent transcription is common at promoters of *C. elegans* genes. GRO-seq and GRO-cap profiles show transcription of a divergent gene pair (**D**) or divergent transcription from a promoter without an upstream divergent gene partner (**E**). Gene on plus strand (red gene and signal). Gene on minus strand (blue gene and signal). (**F**) Upstream divergent transcription from *C. elegans* promoters is intermediate between that in humans and Drosophila. Plot compares the log₂(sense/antisense) transcription ratio of human and fly promoters to *C. elegans* promoters without divergent gene pairs. The median log₂ ratios are 0.3 for humans, 2.3 for *C. elegans*, and 5.0 for Drosophila. RPM: reads per million.

*Figure 4. Continued on next page*

*Figure 4. Continued*

The following source data and figure supplements are available for figure 4:

**Source data 1**. Gene expression, and 5′ pausing and 3′ pausing data for protein-coding genes.

**Figure supplement 1**. The 3′ accumulation of RNA polymerase II.

**Figure supplement 2**. RNA polymerase II accumulation at 3′ ends compared against trans-spliced genes, non-trans-spliced genes, and U-rich regions at 3′ ends.

**Figure supplement 3**. Divergent transcription in *Caenorhabditis elegans*.

promoter to the 3′ end, suggesting that SDC-2 controls sex determination by reducing Pol II recruitment or initiation at the *her-1* promoter in XX embryos (*Figure 5B*).

The median change of gene expression in *sdc-2* mutant vs control embryos was an increase of 1.63- to 1.67-fold for X-linked genes of all lengths and a slight reduction of 0.79- to 0.81-fold for autosome-linked genes (*Figure 5—source data 1* and *Figure 4—source data 1*). For the subset of X and autosomal genes whose expression was statistically different (p≤0.05) between mutant and control embryos, 99% of X-linked genes had elevated expression in the mutants, while 63–66% of autosomal genes had reduced expression (*Figure 5—source data 1* and *Figure 4—source data 1*). The reduced autosomal gene expression was robust to normalization procedures that counteract potential complications from increased X expression (see 'Materials and methods').

Our GRO-seq experiments also provided the first indication that dosage compensation controls the expression of small non-coding RNAs. We found most embryonically expressed X-linked microRNAs to be dosage compensated (*Figure 5D* and *Figure 5—source data 1*, and 'Materials and methods'), while X-linked tRNAs were not, implying that RNA polymerase II is broadly sensitive to dosage compensation, and RNA polymerase III is insensitive (*Figure 5—source data 1*).

Previous studies showed the DCC binds to sequence-specific DNA recruitment sites on X and disperses to promoter regions of actively transcribed genes (*Ercan et al., 2009*; *Jans et al., 2009*; *Pferdehirt et al., 2011*), but the lack of precise TSS calls had prevented an accurate alignment of DCC binding sites with promoters. Mapping of new TSSs relative to DCC binding sites called from our ChIP-seq experiments performed for this comparison showed the peak of DCC binding to be immediately upstream of the TSS (*Figure 5E*). Although the DCC binds to promoter regions, prior studies showed that DCC binding to a gene is neither necessary nor sufficient for the compensation of that gene, and not all genes are dosage compensated (*Jans et al., 2009*). We assessed the generality of that conclusion by comparing DCC binding upstream of a TSS to the increase in gene expression caused by disrupting *sdc-2*. Our results confirmed and extended the original conclusion (*Figure 5—figure supplement 3A,B*). Thus, the DCC can act at a distance to control gene expression, and DCC binding intensity is not a proxy for the dosage compensation status of the gene.

## The *C. elegans* dosage compensation process does not regulate promoter-proximal pausing

To determine the step of transcription controlled by the DCC, we compared the distribution of the GRO-seq signal along X and autosomal genes between control and dosage-compensation-defective embryos. The change in Pol II distribution expected from disrupting dosage compensation differs according to the step of transcription affected by the DCC. If the DCC restricts Pol II recruitment, a uniform increase in engaged Pol II would be expected from the promoter to the 3′ end of genes of mutants. If the DCC reduces transcription by preventing the release of Pol II promoter-proximal pausing, or by reducing transcription elongation at stages downstream of the pause site, an increase in the level of engaged Pol II would be expected in the gene body and 3′ end in mutants, with the increase beginning more promoter proximal for a mechanism that controls pausing.

Promoter-proximal pausing of transcriptionally engaged Pol II is a rate-limiting step of transcription in metazoans (*Adelman and Lis, 2012*) observed at ~ 40% of active genes in mammalian cells (*Core et al., 2008*) and more than 60% of active genes in *Drosophila* (*Core et al., 2012*). To determine whether control of 5′ pausing was a plausible mechanism for dosage compensation, we calculated 5′ pausing ratios (GRO-seq signal for 5′ end/gene body) for genes in each developmental stage using

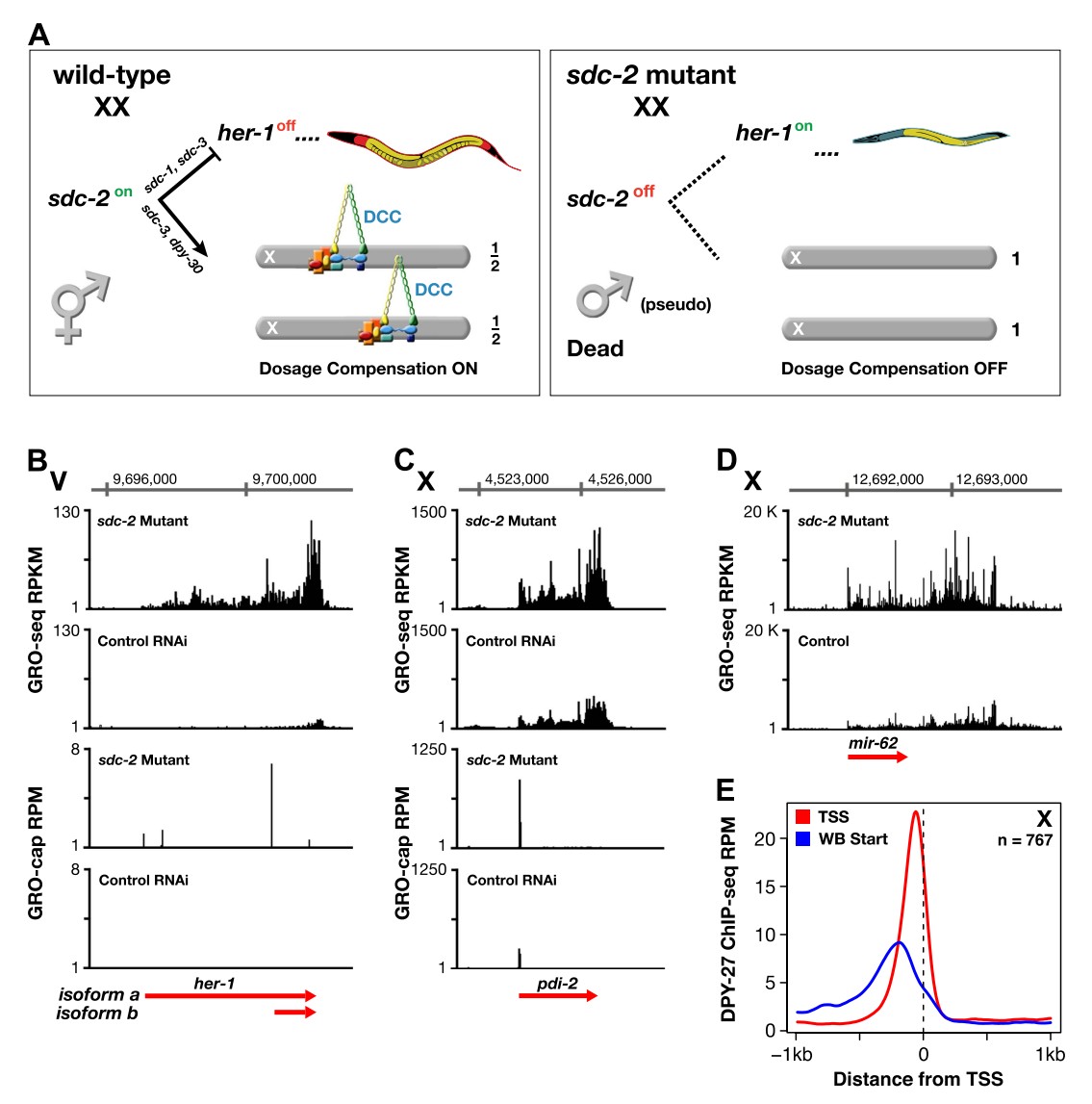

**Figure 5**. GRO-seq analysis of dosage-compensation. (**A**) Genetic hierarchy for coordinate control of sex determination and dosage compensation. *sdc-2* is expressed solely in XX embryos and triggers the hermaphrodite fate. *sdc-2* acts together with *sdc-1* and *sdc-3*, both zinc finger proteins, to induce hermaphrodite sexual development by repressing transcription of the male sex-determining gene *her-1*. *sdc-2* acts together with *sdc-3* and *dpy-30*, also a member of the MLL/COMPASS gene activating complex, to load the DCC onto X and thereby turn dosage compensation on. *sdc-2* is the single gene required for all DCC components to assemble onto X. Without *sdc-2*, *her-1* is expressed, causing sexual transformation of XX embryos to the male fate, and the DCC fails to assemble onto X, causing severe dosage compensation disruption and the death of all XX embryos. The DCC contains not only the X loaders (red, orange) but also five homologs of the mitotic condensin complex (yellow, blue, green). The DCC binds to the X chromosomes of only XX animals to reduce transcription by half, thereby equalizing X-chromosome gene expression between males (XO) and hermaphrodites (XX). (**B**) GRO-seq shows that both RNA isoforms of *her-1* are elevated in *sdc-2* XX mutants. *her-1* is expressed at such a low level in XX embryos that the two transcription start sites (TSSs) are only evident with GRO-cap in the *sdc-2* mutants. The gene model (red arrow) incorporates 3′ end data from ***Jan et al. (2011)***. (**C**) and (**D**) The X-linked protein coding gene *pdi-2* is elevated in expression in *sdc-2* mutants as is the gene encoding the *mir-62* microRNA. For *pdi-2*, the elevation starts at the TSS and is evident throughout the gene. Red arrows show our re-annotated gene models. (**E**) The DCC subunit DPY-27 binds just upstream of the TSS. Comparison of the average DPY-27 ChIP-seq signal relative to WormBase (WB) starts and TSSs of X-linked genes. RPKM: reads per kilobase per million; RPM: reads per million.

The following source data and figure supplements are available for figure 5:

**Source data 1**. Genome-wide changes in gene expression caused by the disruption of dosage compensation.

*Figure 5. Continued on next page*

*Figure 5. Continued*

**Figure supplement 1**. Western blot shows the reduction in SDC-2 protein levels in *sdc-2(y93, RNAi)* animals.

**Figure supplement 2**. X-linked gene expression is selectively increased in *sdc-2* mutants.

**Figure supplement 3**. Occupancy of the DCC subunit DPY-27 in the promoter of a gene is correlated with the gene's expression level but not its dosage compensation status.

TSSs calls for the respective stage (*Figure 4—source data 1*). In contrast to *Drosophila* and mammalian genomes, very few genes (0.38%, 15 of 3975) of wild-type *C. elegans* embryos showed evidence of 5′ pausing (*Figure 4—source data 1*, *Figure 6A,B*, and *Figure 6—figure supplement 1A–D*). The paused genes were not enriched on X (*Figure 6C*). Moreover, the number of 5′ paused genes was not decreased in *sdc-2* mutants (*Figure 4—source data 1* and *Figure 6D,E*).

Nonetheless, the GRO-seq assay is capable of detecting 5′ pausing in *C. elegans*, since genes not paused in embryos become paused in L1s deprived of food (*Figure 6A* and *Figure 6—figure supplement 1A–C*). In starved L1s, we found 7.7% of genes (166 of 2133) to exhibit 5′ pausing, and most were on autosomes (*Figure 4—source data 1*). A prior genome-wide analysis of Pol II binding in L1 animals cultured with and without food discovered an enrichment of Pol II only at some promoters of food-deprived larvae, consistent with 5′ proximal pausing (*Baugh et al., 2009*). Our results confirm their proposal that Pol II pausing can be induced by food deprivation. These cumulative results make it highly implausible that 5′ pausing control is the mechanism of X-chromosome dosage compensation in embryos, and they are consistent with the lack of a *C. elegans* negative elongation factor complex (NELF), which contributes, in part, to pausing in other organisms (*Yamaguchi et al., 1999*).

### The *C. elegans* dosage compensation process regulates the recruitment of Pol II to promoters

Analysis of GRO-seq data comparing transcriptionally engaged Pol II in control vs dosage-compensation-defective XX embryos revealed a uniform increase in the level of engaged Pol II across the entire length of X-linked genes, from TSSs to 3′ ends, in the *sdc-2* mutant. This conclusion was reached both by metagene analysis (*Figure 7A*) and by the analysis of individual genes exhibiting a range of overexpression in the mutant (*Figures 5C and 7C*, and *Figure 7—figure supplement 1*). Analysis of individual genes averts complications in interpretations that might arise from averaging of data.

For the metagene analysis, the ratio of engaged polymerase in mutant vs control was consistently elevated by at least 1.6-fold across scaled X-linked genes, from TSSs to 3′ ends (*Figure 7A*). This elevation in the GRO-seq signal was also readily apparent in metagene analyses of X-linked genes subdivided by quartiles of expression (*Figure 7—figure supplement 2*) or by gene length (*Figure 7—figure supplement 3*). In contrast, metagene analysis of autosomal genes from the same data sets showed only a slight decrease in engaged Pol II across their lengths (*Figure 7B*), consistent with the slight decrease in autosomal gene expression in mutants (*Jans et al., 2009*).

Heat maps displaying the $\log_2$ ratio of mutant to control GRO-seq signal across individual X-linked (*Figure 7C*) and autosomal (*Figure 7D*) genes confirmed the uniform elevation of transcriptionally engaged Pol II across X-linked genes and the reduction across individual autosomal genes of *sdc-2* mutants. Together, these results demonstrate that the dosage compensation mechanism reduces X-linked gene expression in *C. elegans* hermaphrodites by restricting either the recruitment or initiation of Pol II. This conclusion is strongly supported by the analysis of elongation density indices (average GRO-seq signal across the last 75% of genes/average signal across the first 25%) in mutant vs control embryos for genes on X and autosomes. The ratios of indices between control and mutant embryos were not significantly different for genes on X compared to autosomes, indicating that dosage compensation does not preferentially affect Pol II elongation (*Figure 7E*).

Two further lines of evidence show Pol II recruitment to be the predominant step of transcription targeted by the dosage compensation process. First, re-analysis of our previous genome-wide ChIP-chip analysis of Pol II binding (*Pferdehirt et al., 2011*) relative to the new TSSs showed an approximately twofold increase in the occupancy of hypo-phosphorylated Pol II at the promoters of X-linked genes in *sdc-2* mutant vs control embryos (*Figure 7F*). DNA-bound hypo-phosphorylated Pol II at promoters

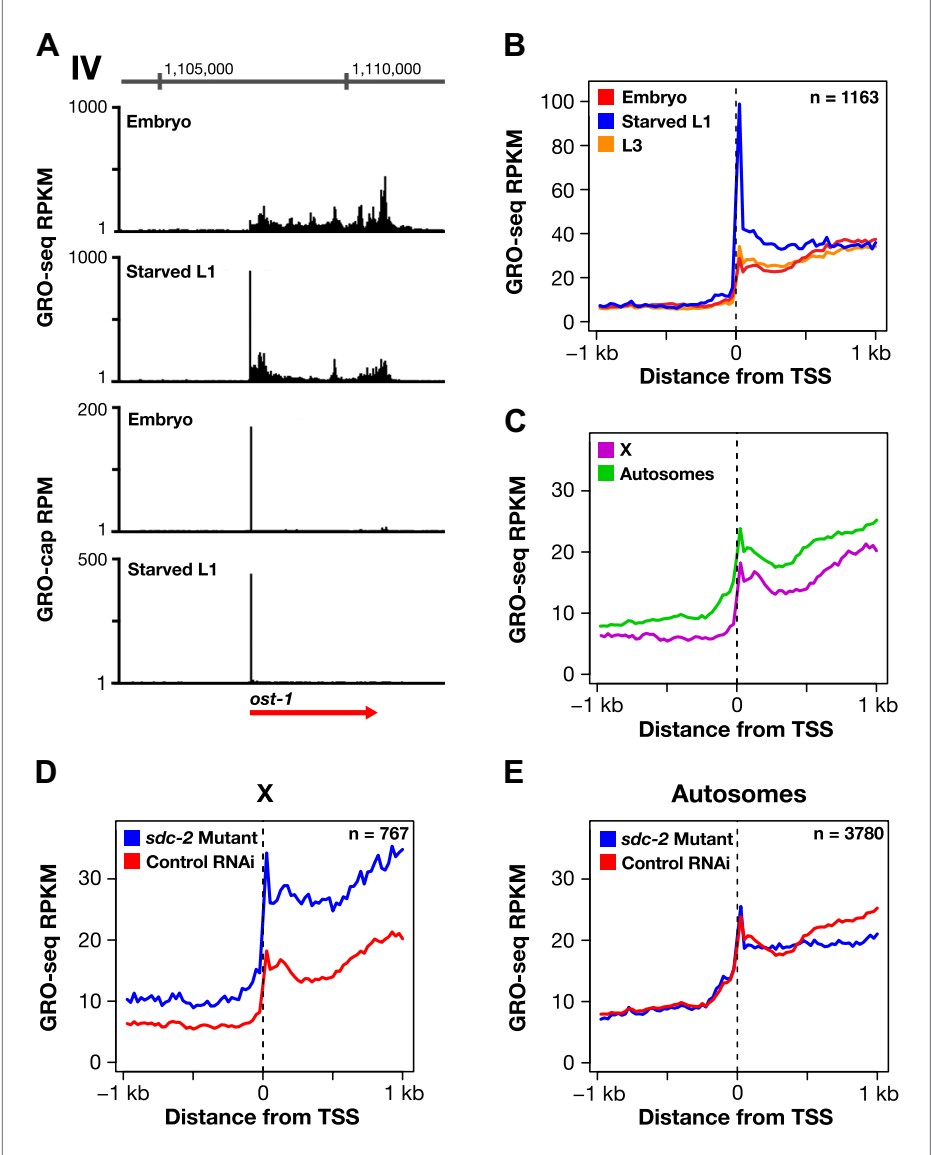

**Figure 6**. Promoter-proximal RNA Pol II pausing is rare in *Caenorhabditis elegans* and is not the target of dosage compensation. (**A**) GRO-seq and GRO-cap signals show that a gene not paused in embryos becomes paused in L1 larvae deprived of food. (**B**) Comparison of average GRO-seq signal from embryos, starved L1 larvae, and L3 larvae within 2 kb of transcription start sites (TSSs) called in all three stages shows that promoter-proximal pausing in embryos and L3s is rare compared to that in starved L1 larvae. (**C**) Promoter-proximal pausing is not enriched on the X chromosome relative to autosomes in embryos. If dosage compensation prevented the release of Pol II from promoter-proximal pause sites, there should be higher levels of pausing on the X chromosome. (**D**) and (**E**) The level of promoter-proximal pausing is not decreased in *sdc-2* mutants compared to control embryos. If dosage compensation reduced gene expression by preventing the release of Pol II from promoter-proximal pause sites, the *sdc-2* mutant should exhibit lower levels of pausing. (**D**) Although X-linked genes have increased expression in *sdc-2* mutants, their level of pausing is not decreased. (**E**) The level of pausing displayed by autosomal genes is unchanged in *sdc-2* mutants. RPKM: reads per kilobase per million; RPM: reads per million.

The following figure supplements are available for figure 6:

**Figure supplement 1**. The dosage compensation process does not control promoter-proximal pausing of Pol II.

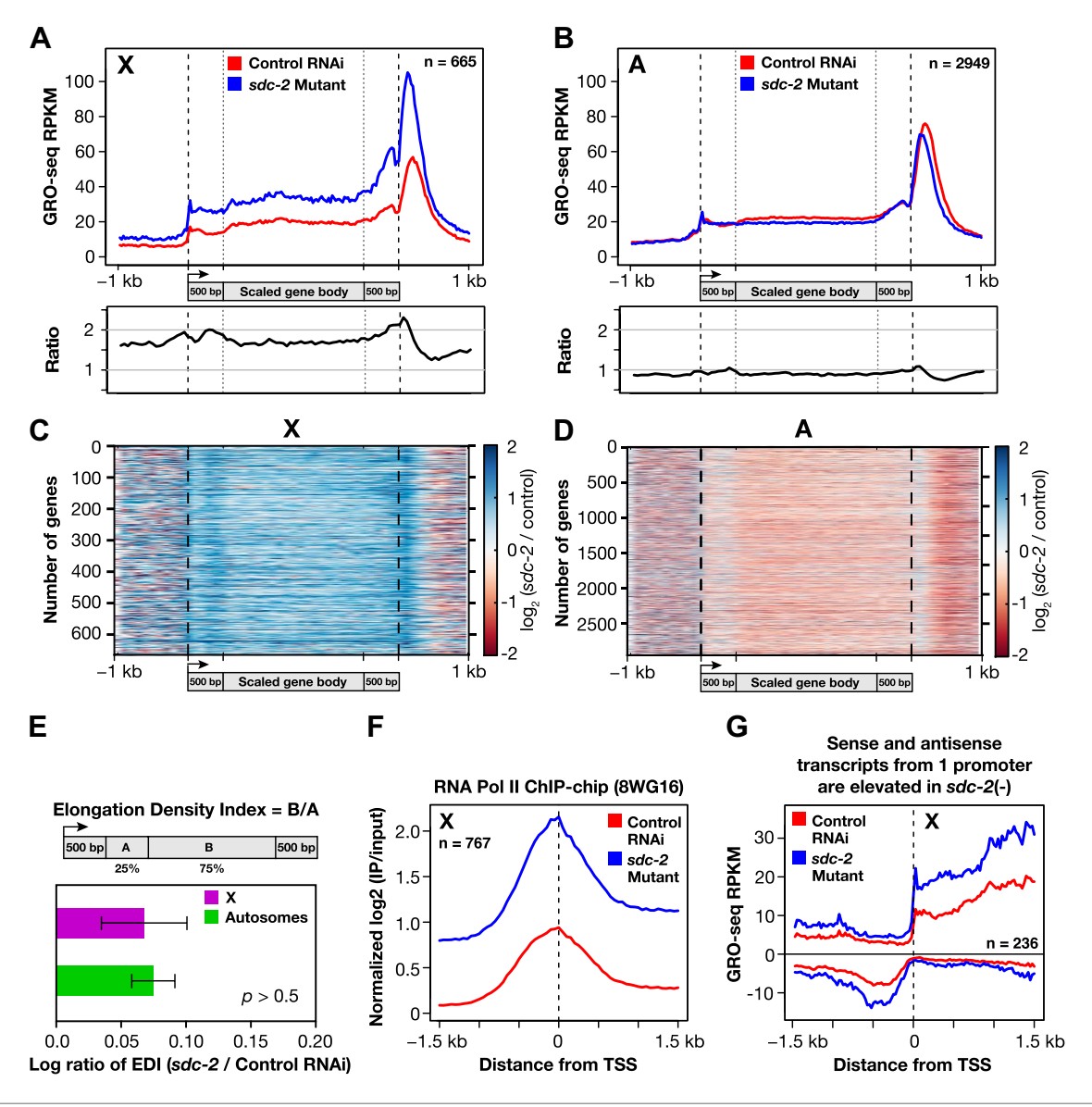

**Figure 7**. The DCC condensin complex reduces X-chromosome gene expression in XX embryos by restricting Pol II recruitment to promoters. (**A**) Uniform increase in GRO-seq signal across the length of X-linked genes results from disrupting dosage compensation. Metagene analysis comparing the average GRO-seq signal from 665 X-linked genes ≥1.5 kb in control RNAi or *sdc-2* mutant embryos. Genes were scaled to the same length as follows: 5' ends (−1 kb to +500 bp of the transcription start site [TSS]) and 3' ends (500 bp upstream to 1 kb downstream of 3' end) were not scaled, and the gene body was scaled to 2 kb. The signal was averaged at each base pair and then averaged across 25 bp windows. The GRO-seq signal is elevated approximately 1.6-fold across genes in *sdc-2* mutant versus control RNAi embryos (below). (**B**) The GRO-seq signal is decreased slightly across autosomal genes in *sdc-2* mutant versus RNAi control embryos. Metagene analysis of 2949 autosomal genes ≥1.5 kb performed as in (**D**). The ratio of the GRO-seq signal in mutant versus control embryos is about 0.9. (**C**) Heat map shows that the GRO-seq signal is increased along the length of each X-linked gene in *sdc-2* mutants. For each of 665 genes, the GRO-seq signal from mutant or control embryos was summed across 100 bp windows and the *sdc-2*/control ratio was calculated for each window. The $\log_2$(*sdc-2*/control) value was plotted across the scaled gene. (**D**) Heat map shows that the GRO-seq signal is moderately decreased along the length of individual autosomal genes in *sdc-2* mutants. For each of 2949 autosomal genes, the $\log_2$(*sdc-2*/control) value was plotted across the scaled gene as in (**F**). (**E**) Dosage compensation does not specifically affect Pol II elongation. An elongation density index was calculated for each gene greater than 2 kb in length that did not have another gene on the same strand within 1 kb of the TSS. After excluding the first and last 500 bp of the gene, the average signal across the last 75% of the remaining gene was divided by the average signal across the first 25% of the remaining gene. Ratios of the indices between the *sdc-2* mutant and control RNAi embryos are not significantly different for genes on the X compared to the autosomes. Error bars represent a 95% confidence interval for the mean indices. n = 481 (X); n = 1861 (autosomes). (**F**) Occupancy of hypo-phosphorylated Pol II at the promoters of X-linked genes is increased in dosage compensation mutants, showing

*Figure 7. Continued on next page*

*Figure 7. Continued*

greater Pol II recruitment. Comparison of normalized Pol II ChIP-chip signal from control RNAi or *sdc-2* mutant embryos relative to newly annotated TSSs of X-linked genes. (**G**) Sense and upstream divergent transcription are coordinately increased for X-linked genes in *sdc-2* mutants. Comparison of average sense or antisense GRO-seq signal from *sdc-2* mutant and control RNAi embryos for a 3 kb window surrounding TSSs for genes with no divergent gene partner. The GRO-cap signal was only evaluated in this analysis for genes having a log$_2$(sense/antisense) ratio ≤1.5 in control RNAi embryos.

The following figure supplements are available for figure 7:

**Figure supplement 1**. GRO-seq signal is increased along the length of individual X-linked genes when dosage compensation is disrupted.

**Figure supplement 2**. Disruption of dosage compensation causes a uniform increase in GRO-seq signal across the length of X-linked genes in different quartiles of gene expression determined from control RNAi samples.

**Figure supplement 3**. Disruption of dosage compensation causes a uniform increase in GRO-seq signal across the length of X-linked genes of different size ranges.

**Figure supplement 4**. The level of antisense transcription is unaffected by dosage compensation.

is enriched for non-initiated Pol II. The increase in Pol II occupancy at promoters of mutants (measured by ChIP) was nearly equivalent to the increase in post-initiated Pol II (measured by GRO-seq), indicating that Pol II promoter recruitment is rate-limiting when dosage compensation is active.

Second, in *sdc-2* mutants, the levels of sense and upstream divergent transcription were elevated coordinately at promoters on X lacking upstream divergent genes (*Figure 7G* and *Figure 7—figure supplement 4A*). At those promoters, the level of hypo-phosphorylated Pol II was also elevated about twofold in mutant vs control embryos, as assayed by ChIP (*Figure 7—figure supplement 4B*). Since we found increased Pol II occupancy in mutants and observed virtually no promoter-proximal pausing in control embryos, escape from pausing into productive elongation, or any other form of post-initiation regulation, cannot be the cause of elevated transcription at the divergent promoters in *sdc-2* mutants. Furthermore, an increase in transcription initiation by bound polymerases cannot account for these results, but an increase in Pol II recruitment can. Our combined experiments demonstrate that *C. elegans* dosage compensation controls X-chromosome-wide gene expression predominantly by reducing Pol II recruitment to promoters of hermaphrodites.

## *C. elegans* balances gene expression between X chromosomes and autosomes

In organisms that equalize X-chromosome gene expression between the sexes by a dosage compensation process, the question arises as to whether the compensated level of X-chromosome expression is equivalent to or half of the expression from the two sets of autosomes. The answer to this question has been controversial, although evidence has mounted in favor of a mechanism to balance total expression between X chromosomes and the two sets of autosomes based on measurements of accumulated transcripts (*Xiong et al., 2010*; *Deng et al., 2011*; *Disteche, 2012*; *Lin et al., 2012*; *Deng et al., 2013*; *Jue et al., 2013*). Our GRO-seq experiments have addressed this question in the most definitive way to date by measuring the levels of nascent transcripts prior to co-transcriptional processing. We show that in wild-type *C. elegans* embryos, X and autosomes have nearly equivalent levels of total gene expression, and that levels of transcribing Pol II are uniform across X and autosomal genes (*Figure 8A*). In dosage-compensation-defective mutants, the level of X expression and engaged Pol II exceeds that of autosomes by 1.7-fold, from the TSSs to the 3′ ends (*Figure 8B*). These results demonstrate the existence of a separate mechanism in *C. elegans* to elevate the intrinsic rate of transcription from the X chromosomes of both sexes, so that after dosage compensation, X chromosomes and the two sets of autosomes have equivalent expression. Our experiments provide evidence that it, like dosage compensation, works at the level of controlling Pol II recruitment.

## Discussion

A critical step in dissecting the mechanism of dosage compensation in any organism is to determine the primary step of transcription controlled by the dosage compensation complex to balance X-chromosome gene expression between the sexes. A major obstacle towards attaining this goal for

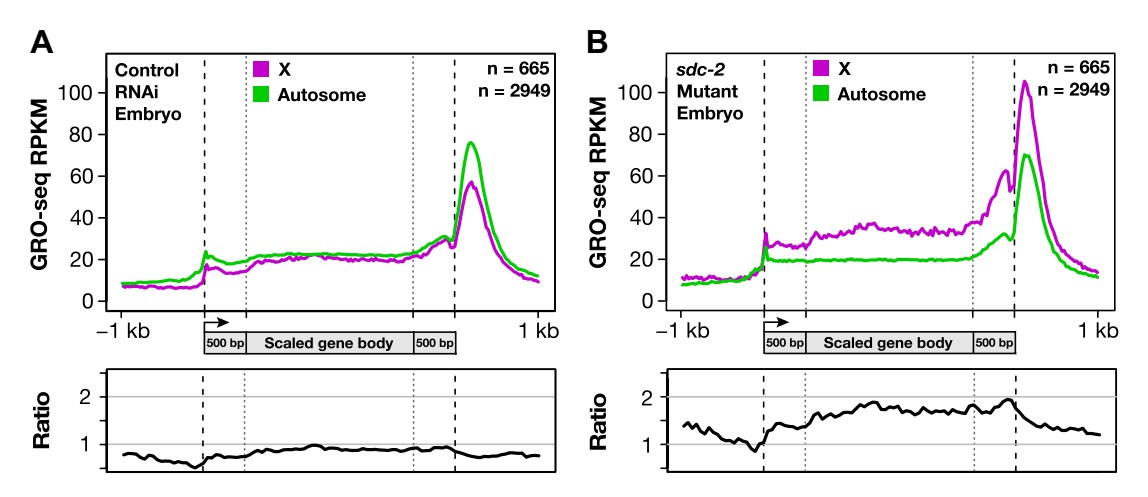

**Figure 8**. Gene expression is balanced between X chromosomes and autosomes. (**A**) *Caenorhabditis elegans* has a mechanism to equalize expression between X chromosomes and autosomes. Metagene analysis comparing the average GRO-seq signal from X-linked and autosome-linked genes of control RNAi embryos. The X to autosome expression ratio is 0.9. (**B**) In dosage-compensation-defective mutants, the level of X-chromosome expression exceeds that of autosomes by 1.7-fold. Metagene analysis comparing the average GRO-seq signal from X-linked and autosome-linked genes of *sdc-2* mutant embryos. RPKM: reads per kilobase per million.

*C. elegans* had been the lack of properly annotated transcription start sites for the majority of its genes. We therefore initiated our studies by developing a general and efficient strategy to map TSSs for Pol II-regulated genes with nucleotide resolution. This method, called GRO-cap, utilized purified nascent transcripts from the NRO reactions developed for GRO-seq and employed redundant enzymatic steps to capture RNAs with 5′ caps. GRO-cap tracks RNAs prior to their co-transcriptional processing. This approach was critical for our analysis because co-transcriptional processing removes TSSs from the transcripts of most nematode genes, including protein-coding genes and small non-coding RNA genes.

We found TSSs for protein-coding genes to be unexpectedly far upstream from the 5′ ends of mature mRNAs. True TSSs were as far as 14 kb upstream of previously annotated 5′ ends (typically the trans-spliced acceptor sites), and 59% of TSSs were >200 bp upstream and 21% >1 kb upstream. GRO-cap was also effective at identifying TSSs for the majority of small processed RNAs, including microRNAs, tRNAs, snoRNAs, and 21 U-RNAs.

To determine the step of transcription controlled by the DCC, we compared the genome-wide distribution of transcriptionally engaged Pol II (from GRO-seq) relative to the newly defined TSSs (from GRO-cap) in wild-type vs dosage-compensation-defective XX embryos. We found 5′ promoter-proximal pausing to be rare, unlike in other metazoans, and not likely to be the target of dosage compensation. Instead, dosage compensation reduces recruitment of RNA Pol II to the promoters of hermaphrodite X-linked genes, thereby decreasing transcription from the two X chromosomes of hermaphrodites to the level of transcription from the single X chromosome of males. A separate regulatory mechanism elevates the intrinsic level of transcription from X chromosomes of both sexes so that after dosage compensation total expression from X chromosomes and autosomes is equivalent.

## Advantages of GRO-cap

GRO-cap has advantages over TSS mapping strategies that use total accumulated RNA or short-capped (sc) RNA as the starting material. The use of nascent RNAs without base hydrolysis or size selection, as in GRO-cap, enriches the proportion of 5′ capped RNAs within the starting RNA population, reduces the level of false TSS calls from RNAs that are capped post-transcriptionally (*Fejes-Toth et al., 2009*), and increases the probability of identifying TSSs from promoters that are transcribed at low levels. Most importantly, the GRO-cap strategy permits reliable assignment of TSSs by pairing TSS calls with uninterrupted GRO-seq signal for transcriptionally engaged Pol II between the GRO-cap TSS and the previously annotated 5′ end. Multiple lines of evidence showed that GRO-cap signal combined with continuous GRO-seq signal from WB starts defines true TSSs.

Two reports of nematode TSSs have recently been published. The first used two approaches to identify the transcription starts (*Gu et al., 2012*). The first, called CapSeq, used total accumulated RNA as the starting material for the enzymatic enrichment of RNAs with 5′ caps and set the cutoff for TSS calls to be within 1 kb upstream and 100 bp downstream from previously annotated 5′ ends. As a consequence, CapSeq did not identify TSSs for a large class of protein-coding genes. TSSs of small processed non-coding RNAs were also difficult to identify by CapSeq. The second approach, called CIP-TAP, used scRNAs and was equivalently effective as GRO-cap for identifying TSSs of small non-coding RNAs.

The second report also used scRNAs to map TSSs and set the cutoff for TSS calls to be no greater then 200 bp upstream of the previously annotated 5′ ends, thereby also not defining TSSs for a large class of genes (*Chen et al., 2013*). The authors found numerous clusters of Pol II initiation upstream of their calls and classified many of these initiation events as enhancer-like chromatin signatures based on overlap with bound transcription factors. Based on comparison with our data (e.g., *Figure 1—figure supplement 12A,B*), we propose that a proportion of the upstream enhancer-like signatures are TSSs giving rise to full-length transcripts. Two clear examples of genes are shown in *Figure 1—figure supplement 12B,C*, where the single TSS (either 2534 bp or 2878 bp upstream of the WB start) called from GRO-cap and GRO-seq data was classified as an enhancer by *Chen et al. (2013)*.

## Features of *C. elegans* transcription

Although 5′ promoter-proximal pausing in metazoans is a common rate-limiting step of transcription that is highly regulated to control gene expression (*Adelman and Lis, 2012*), we found it to be rare in *C. elegans* under normal growth conditions. However, in L1 larvae deprived of food, we found 5′ pausing at 7.7% of genes with TSS calls. A prior study using ChIP-seq discovered the accumulation of Pol II at promoters of starved larvae and proposed that 5′ promoter-proximal pausing was responsible for Pol II accumulation (*Baugh et al., 2009*). Our results confirm that interpretation. Two factors promote 5′ pausing in metazoans, negative elongation factor (NELF) and DRB sensitivity-inducing factor (DSIF), although the relative contribution of each is not fully understood (*Adelman and Lis, 2012*; *Yamaguchi et al., 2013*). *C. elegans* appears to lack NELF, suggesting either that DSIF is sufficient in the infrequent cases of pausing or that the core promoter complex (*Kwak et al., 2013*) or another not-yet identified negative elongation factor participates.

The 3′ accumulation of Pol II has been documented previously in flies and humans using GRO-seq (*Core et al., 2008*, *2012*). We found 3′ Pol II accumulation to be more extensive in nematodes than flies. Pol II 3′ accumulation is likely caused by slow 3′ end formation and RNA processing (*Gromak et al., 2006*). Because *C. elegans* has numerous polycistronic operons requiring extensive RNA processing to produce monocistronic mRNAs (*Blumenthal, 2012*), we were able to discover a strong positive correlation between the amount of 3′ Pol II accumulation and the amount of RNA processing. Curiously, though, even genes that were not part of operons exhibited greater 3′ accumulation than genes in flies. This observation raises the question of whether RNA processing in *C. elegans* is generally slower than that in other organisms to accommodate extensive trans-splicing, or whether *C. elegans* accommodates operons by imposing additional regulation on Pol II to enhance its 3′ pausing at all genes to assess whether to halt transcription or continue elongation.

## *C. elegans* dosage compensation controls RNA Pol II recruitment

Gene expression in metazoans is controlled by diverse regulatory mechanisms that function over widely different distances. Some mechanisms act locally on individual genes, while others such as dosage compensation function across large chromosomal territories or along entire chromosomes to regulate a large set of genes coordinately. In general, the step of transcription controlled by long-range mechanisms is not understood. In our study, multiple lines of evidence supported the conclusion that *C. elegans* dosage compensation regulates gene expression along X primarily by reducing the recruitment of RNA Pol II to the promoters of hermaphrodite X-linked genes.

First, regulation of 5′ promoter-proximal pausing cannot be the mechanism underlying *C. elegans* dosage compensation. If the DCC reduced X-chromosome gene expression by increasing 5′ promoter-proximal pausing, numerous genes on X would be paused in wild-type embryos, and disruption of dosage compensation would reduce the level of pausing. GRO-seq experiments showed instead that 5′ promoter-proximal pausing is rare in wild-type XX embryos. The few genes that exhibited 5′ pausing were not enriched on X chromosomes relative to autosomes, and the level of pausing was not decreased in dosage-compensation defective mutants.

Second, Pol II-mediated transcription elongation is not preferentially affected by the DCC. The ratios of elongation density indices (average GRO-seq signal in last 75% of a gene/average GRO-seq signal in first 25% of a gene) calculated for genes on X chromosomes and autosomes in control vs dosage-compensation-defective embryos were not significantly different between X-linked and autosomal genes, indicating that Pol II-mediated transcription elongation is not selectively changed on X chromosomes of mutants.

Third, the level of transcriptionally engaged Pol II assayed genome-wide by GRO-seq in control vs dosage-compensation-defective XX embryos revealed a uniform increase in engaged Pol II across the entire length of X-chromosome genes, but not autosomal genes, in mutants. Hence, the DCC controls X-chromosome gene expression in XX animals by reducing either the recruitment or initiation of Pol II. This conclusion was validated both by metagene analysis and by analysis of hundreds of individual X-linked genes showing different levels of de-repression in dosage-compensation-defective mutants.

Fourth, genome-wide quantification of Pol II occupancy by ChIP plotted relative to the new TSSs showed an increase in the hypo-phosphorylated form of Pol II at promoters of dosage-compensation-defective embryos vs control embryos that was equivalent to the increase in post-initiated Pol II measured by GRO-seq. Since DNA-bound hypo-phosphorylated Pol II at promoters is enriched for non-initiated Pol II, these results indicate that Pol II promoter recruitment is rate limiting when dosage compensation is active.

Our combined experiments reveal that the primary mechanism by which the *C. elegans* dosage compensation process reduces X-chromosome gene expression by half in XX embryos is to limit Pol II recruitment to promoters of X-linked genes. Our study does not eliminate the possibility of a minor repressive influence acting through another step of transcription or through a post-transcriptional mechanism such as RNA stability.

## DCC function

How might condensin reduce Pol II recruitment to the promoters of X-linked genes by approximately twofold? Our current and prior (*Jans et al., 2009*) studies showed that DCC binding to the promoter of a gene is neither necessary nor sufficient to elicit repression of the gene. Hence, the DCC influences gene expression over long distance, likely by imposing changes in higher-order chromosome structure. Clues to such a DCC function were suggested originally by the simultaneous discovery of the DCC and condensin's biochemical properties in vitro as an ATPase that alters DNA topology (*Chuang et al., 1994*; *Kimura and Hirano, 1997*; *Hagstrom et al., 2002*; *Hirano, 2012*; *Piazza et al., 2013*) and its canonical roles in vivo of compacting and resolving mitotic and meiotic chromosomes for proper chromosome segregation (*Hirano and Mitchison, 1994*; *Hagstrom et al., 2002*; *Chan et al., 2004*; *Hirano, 2012*; *Piazza et al., 2013*). However, mechanisms of DCC function are perhaps best deduced from its non-canonical roles in vivo of regulating interphase chromosome structure (*Bauer et al., 2012*) and meiotic crossover recombination (*Mets and Meyer, 2009*; *Wood et al., 2010*; *Aragon et al., 2013*). Condensin II in *Drosophila* induces axial compaction of interphase chromosomes, globally disrupts inter-chromosomal interactions, and promotes dispersal of peri-centric heterochromatin (*Bauer et al., 2012*). These activities serve to compartmentalize the interphase nucleus into discrete chromosomal territories. Furthermore, nematodes carrying mutations that disrupt condensin I, which shares four subunits with the DCC, display elongated chromosomal axes during meiotic prophase and exhibit chromosome-wide changes in the distribution of double strand breaks and crossovers (*Mets and Meyer, 2009*), providing strong evidence that DCC subunits control intra-chromosomal structure.

These cytological, biochemical, and genetic observations of condensin function suggest that chromosome structure might affect Pol II recruitment in several ways, which are not mutually exclusive. Chromosome compaction and associated topological changes in DNA could broadly affect promoter accessibility of Pol II and the regulatory factors that recruit or stabilize it, thereby reducing Pol II recruitment in a quantitatively similar manner at different sites. Compaction of interphase chromosomal territories could reduce the local concentration of bound transcription activators and hence bound Pol II. Changes in intra-chromosomal interactions could alter the relationships between distal regulatory regions and their target promoters, thereby limiting Pol II recruitment. These and other models are topics of future studies.

## Dosage compensation mechanisms differ between nematodes and flies

Dosage compensation strategies differ across species. Mammals inactivate one of the two female X chromosomes, flies double expression of the single male X chromosome, and nematodes halve

expression of both hermaphrodite X chromosomes. A central question is whether the molecular mechanisms underlying these diverse forms of chromosome-wide transcriptional regulation are the same or different.

In *Drosophila melanogaster*, dosage compensation is achieved by the male-specific lethal (MSL) complex that binds along the single X chromosome of males to double transcription of X-linked genes (*Gelbart and Kuroda, 2009*; *Conrad and Akhtar, 2012*). The complex contains two long non-coding RNAs and five proteins, including the H4K16 histone acetyltransferase MOF (*Hilfiker et al., 1997*; *Conrad et al., 2012a*) and the H2BK34 ubiquitin ligase MSL2 (*Wu et al., 2011*). The MSL complex was proposed to regulate gene expression by controlling transcription elongation (*Smith et al., 2001*). This model was subsequently supported by genome-wide mapping of MSL proteins and MSL-dependent H4K16ac to the bodies of male X-linked genes, with a bias toward 3' ends (*Alekseyenko et al., 2006*; *Gilfillan et al., 2006*; *Legube et al., 2006*), and GRO-seq experiments in S2 cells with or without RNAi of the *msl2* gene (*Larschan et al., 2011*). While a recent study analyzing genome-wide Pol II occupancy in males and females by ChIP-seq experiments suggested an alternative view that fly dosage compensation operates at the level of Pol II recruitment or initiation (*Conrad et al., 2012b*), a mathematical computation error discovered in this study (*Ferrari et al., 2013*; *Straub and Becker, 2013*) rendered the results insufficient to distinguish between the competing models. The elongation model received recent further support from the discovery that dosage compensation is disrupted by impairing the function of SPT5, a transcription elongation factor that co-localizes with the MSL complex on male X chromosomes and interacts physically with the MSL complex (*Prabhakaran and Kelley, 2012*). Thus, the weight of evidence strongly favors enhancement of transcription elongation as the primary mechanism of fly dosage compensation.

Not only does the overall dosage compensation strategy differ between worms and flies, the underlying molecular mechanism appears to differ as well. In worms, reduction in X-chromosome gene expression is primarily achieved by reducing recruitment of Pol II to promoters, while in flies, elevation in X-chromosome gene expression is primarily achieved by facilitating Pol II transcription elongation. Multiple solutions have evolved to coordinately control gene expression across an entire chromosome.

## Balancing gene expression between X chromosomes and autosomes

In many species, the evolution of sex chromosomes to be the primary determinants of sexual fate resulted in males having one X chromosome and females having two. Such chromosome sex-determining mechanisms had the potential to cause two problems in gene expression, an imbalance in X-chromosome gene expression between the sexes and an imbalance in gene expression between the male X chromosome and his two sets of autosomes (*Disteche, 2012*). X-chromosome dosage compensation strategies such as the *Drosophila* strategy solved both problems by doubling transcription of the single male X chromosome. An alternative solution would be to co-evolve two mechanisms, one to increase expression of X chromosomes in both sexes, thereby preventing hypo X expression in males, and a second to decrease total expression from female X chromosomes to prevent hyper X expression relative to female autosomes and male X chromosomes. Nematodes and mammals evolved the second strategy for equalizing X-chromosome expression between the sexes. It has been controversial whether these organisms also evolved a strategy to elevate X expression in both sexes and thereby balance expression between X chromosomes and autosomes (*Xiong et al., 2010*; *Deng et al., 2011*; *Disteche, 2012*; *Lin et al., 2012*; *Deng et al., 2013*; *Jue et al., 2013*).

The bulk of evidence now favors the presence of a mechanism to up-regulate X-chromosome expression in males and females of both organisms (*Deng et al., 2011, 2013*; *Jue et al., 2013*), and our results provide the most compelling evidence to date for *C. elegans.* Our approach of examining nascent transcripts in mixed-stage embryos by GRO-seq conferred three advantages. First, it enabled us to quantify transcription specifically from somatic embryonic cells and thereby avert the complication of quantifying maternally contributed germline transcripts that contaminate mature embryo mRNA. Second, we could examine nascent transcription at a stage for which most germline-specific mechanisms of gene regulation would have been erased. Third, we could quantify transcriptionally engaged Pol II across the entire length of a gene, starting from the bona fide TSS, and assess the step of transcription controlled by the process. We found that a major part of the mechanism to increase X expression in *C. elegans* is to increase the level of Pol II recruitment to promoters, as was shown recently for mammals (*Deng et al., 2013*). Together, these results reinforce the evolutionary importance of balancing gene expression across all chromosomes of a genome.

# Materials and methods

## Nematode strains

The following nematode strains were used:

- Wild-type strain N2
- *sdc-2* mutant, *sdc-2(y93,RNAi)*
- Control RNAi, N2(L4440 *RNAi*).

## Worm growth

Bacteria for use in feeding RNAi were prepared by growing HT115 bacteria bearing an RNAi plasmid (*sdc-2* or the L4440 negative control) overnight in TB and ampicillin, inducing for 2 hr with 1 mM IPTG, pelleting, and resuspending in 1 vol (wt/vol) of LB with 20% glycerol. For RNAi treatment, embryos were harvested from gravid hermaphrodites and allowed to hatch off for 24 hr. NG agar plates supplemented with 1 mM IPTG and 100 µg/ml carbenicillin were spotted with super-induced RNAi bacteria and allowed to induce further overnight at 25°C. Hatched off L1 larvae were spotted on the RNAi plates and grown at 20°C until gravid.

For wild-type samples, embryos were harvested from gravid hermaphrodites grown at 20°C on concentrated HB101. Embryos were allowed to hatch off and starve for 24 hr at 20°C. Worms were fed and grown to L3 stage under liquid culture (1 worm/µl; 10 mg/ml HB101) for 34 hr at 20°C.

## Isolation of nuclei

Animals were collected from whichever stage was desired. After washing twice with M9 buffer, animals were washed with cold nuclear isolation buffer (250 mM sucrose, 10 mM Tris-HCl (pH 7.9), 10 mM $MgCl_2$, 1 mM EGTA, 0.25% NP-40, 1 mM DTT, protease inhibitors, 4 U/ml SUPERaseIn [AM2696; Ambion, Grand Island, NY]). Animals were resuspended in nuclear isolation buffer (embryos and starved L1 in 3 vol, L3 in 1 vol), and dripped into liquid nitrogen to freeze. Starved L1 and L3 samples were ground under liquid nitrogen by mortar and pestle. Larval samples, post-grinding, and embryo samples were dounced with a Kontes 2 ml glass dounce to release nuclei. Douncing and collection of nuclei was performed for up to six rounds as follows: dounce with 10X pestle A, 10X pestle B, 5 min centrifugation at 100×*g*, removal of nuclei-containing supernatant, and addition of an equal volume of nuclear isolation buffer to the pellet. Nuclear isolation was monitored each round to determine effectiveness and when it was complete. The pooled supernatant was centrifuged for 5 min at 1000×*g* to pellet nuclei. The nuclear pellet was washed with nuclear freezing buffer (40% glycerol, 50 mM Tris-HCl (pH 8.3), 0.1 mM EDTA, 5 mM $MgCl_2$, 1 mM DTT, protease inhibitors, 4 U/ml SUPERaseIn). Approximately $1 \times 10^8$ nuclei were resuspended in 100 µl nuclear freezing buffer and stored at −80°C until GRO-seq reactions were performed.

## Preparation of GRO-seq libraries

### NRO reaction

Nuclei (100 µl) were mixed with an equal volume of reaction buffer (10 mM Tris-HCl (pH 8.0), 5 mM $MgCl_2$, 1 mM DTT, 300 mM KCl, 20 U of SUPERaseIn, 1% sarkosyl, 500 µM each of ATP, GTP, and Br-UTP, 2 µM CTP, and 0.33 µM α-$^{32}$P-CTP [3000 Ci/mmol]). The reaction was allowed to proceed for 5 min at 30°C. The reaction was stopped by the addition of 2 ml (10× volume) of TRIzol (Invitrogen). The phases were separated by the addition of 400 µl of chloroform as per the manufacturer's instructions. An additional acid-phenol and then chloroform extraction were carried out, followed by precipitation with 2.5 vol of ethanol. The pellet was washed in 75% ethanol before resuspending in 20 µl of DEPC-treated water. Base hydrolysis was performed on ice by the addition of 5 µl 1 M NaOH and incubated on ice for 30 min. The reaction was neutralized by the addition of 25 µl 1 M Tris-HCl (pH 6.8). The reaction was then run through a p-30 RNAse-free spin column (BioR, Hercules, CA) according to the manufacturer's instructions. The column flowthrough was brought to 100 µl with DEPC water and EDTA was added to a final concentration of 1 mM.

### Bead pre-wash

All buffers used in bead enrichment steps were kept on ice and were supplemented with 4 U/ml of SUPERaseIn. Anti-deoxyBrU beads (#sc-32323-ac; Santa Cruz Biotech, Santa Cruz, CA) were first washed three times with a pre-wash buffer: 0.25× SSPE, 500 mM NaCl, 1 mM EDTA, 0.05% Tween for 5 min;

washed twice in binding buffer: 0.25× SSPE, 37.5 mM NaCl, 1 mM EDTA, 0.05% Tween for 5 min; blocked in bead blocking buffer: 0.25× SSPE, 1 mM EDTA, 0.05% Tween, 0.1% PVP, and 1 mg/ml ultrapure BSA (AM2618; Ambion) for 1 hr; followed by one wash in binding buffer for 5 min. The ratio of beads to volume did not exceed 1:8 for any wash or blocking step. The beads were resuspended in a 25% slurry (original concentration).

### Bead enrichment
NRO RNA was heat denatured at 70°C for 3 min and placed on ice for 2 min. Then, 350 µl of binding buffer and 50 µl of bead slurry were added to the RNA, and the samples were incubated for 30 min on a rotating stand (8 rpm). The beads were washed once in binding buffer; once in low salt wash buffer: 0.2× SSPE, 1 mM EDTA, 0.05% Tween; once in high salt wash buffer: 0.25% SSPE, 137.5 mM NaCl, 1 mM EDTA, 0.05% Tween; and twice in TET: 10 mM Tris-HCl (pH 7.5), 1 mM EDTA, 0.05% Tween. The NRO RNA was eluted three times (2× 125 µl, 1× 250 µl) with elution buffer: 20 mM DTT, 150 mM NaCl, 5 mM Tris-HCl (pH 7.5), 1 mM EDTA, and 0.1% SDS. The NRO RNA was then isolated by a standard extraction-precipitation method: one acid-phenol extraction, one chloroform extraction, addition of NaCl to 300 mM and 1 µl of glycoblue (AM9515; Ambion) to the aqueous phase, precipitation with 2.5 vol of cold ethanol, and a wash of the resulting pellet with 75% ethanol. The pellet was resuspended in DEPC water at volumes appropriate for the subsequent step.

### 3′ End repair
NRO RNA was heated 70°C for 3 min, followed by incubation on ice for 2 min. Then 3 µl 10× T4 PNK buffer, 1 µl SUPERaseIn, and 2 µl of T4 polynucleotide kinase (NEB, Ipswich, MA) were added and the reaction incubated for 30 min at 37°C. The reaction was brought to 100 µl with DEPC water and EDTA to a final concentration of 10 mM to stop the reaction. RNA was heat denatured as above and subjected to two more rounds of bead binding/elution as described above.

### Poly-A tailing, reverse transcription, and amplification
NRO RNAs were then polyadenylated as described in *Ingolia (2010)*. Poly-A tailed RNA (20 µl) was then reverse transcribed as follows: 2 pmol of reverse transcription primer (TruSeq_circ_RTP_pT: 5′-/5Phos/GATCGTCGGACTGTAGAACTCTGAAC/iSP18/CACTCA/iSP18/GCCTTGGCACCCG AGAATTCCATTTTTTTTTTTTTTTTTTTTTTVN-3′), 1 µl 12.5 mM dNTPs, and 3 µl of DEPC water were added the mixture incubated at 75°C for 3 min. The primer (note the low concentration) is allowed to anneal at 42° for 10 min. Then, 1 µl 0.1 mM DTT, 1 µl 1 M Tris-HCl (pH 8.3), 1 µl SUPERaseIN, and 1 µl SuperScript III reverse transcriptase (18080051; Invitrogen) were added and the reverse transcription was carried out at 48°C for 5 min and 54°C for 30 min. The reaction was stopped by incubation at 70°C for 10 min. The reaction was extracted once with buffer saturated phenol–chloroform (pH 8.0), once with chloroform, supplemented with 2 µl glycoblue and 300 mM NaCl, and then precipitated with 2.5 vol of ethanol.

The NRO cDNA library was then PAGE purified away from excess RT primer on an 8% denaturing PAGE gel as described (*Ingolia 2010*). The purified library was then circularized in an intra-molecular ligation reaction as follows: cDNA was resuspended in 15 µl of 10 mM Tris-HCl (pH 8.0) and mixed with 2 µl of 10× CircLigase buffer, 1 µl 1 mM ATP, 1 µl of 50 mM $MnCl_2$ and 1 µl of CircLigase (Epicentre, Madison, WI). The reaction was incubated at 60°C for 90 min, with a brief centrifugation every 15 min to bring down condensation. The reaction was heat inactivated at 80°C for 10 min, extracted once with buffer saturated phenol–chloroform (pH 8.0), once with chloroform, supplemented with 2 µl glycoblue and 300 mM NaCl, and then precipitated with 2.5 vol of ethanol.

The circularized NRO cDNA library was resuspended in 10 µl of water and amplified and PAGE purified as described (*Core et al., 2008*) and quantified before submission for sequencing. Amplification was achieved using the TruSeq miRNA cloning oligo set (RP1: 5′-AATGATACGGCGACCACCGAGAT CTACACGTTCAGAGTTCTACAGTCCGA-3′ [Illumina part # 15005505] and RPI-#: 5′-CAAGCAGAA GACGGCATACGAGATNNNNNNGTGACTGGAGTTCCTTGGCACCCGAGAATTCCA-3′, where the '#' represents one of the 48 Illumina six-base barcodes in the middle of the oligo, shown above as 'NNNNNN').

## Preparation of GRO-cap libraries
All NRO reactions and bead enrichment steps for GRO-cap were carried out as described above, with the exception that the RNA for GRO-cap was not base hydrolyzed. After the first bead binding, the TruSeq RNA 3′ Adapter (RA3) (Illumina part # 15013207) was ligated to the 3′ end of the NRO RNA.

First, 50 pmol of the 3′ adapter were mixed with NRO RNA and 2 µl 50% PEG 8000, brought to 14 µl with DEPC water, incubated at 70°C for 3 min and put on ice for 2 min. Then, 2 µl of 10× T4 RNA Ligase I buffer, 2 µl 10 mM ATP, 1 µl SUPERaseIN, and 1.5 µl T4 RNA Ligase I (M0204; NEB) were added and reaction incubated at 22°C for 4–6 hr. The reaction was then brought to 100 µl with binding buffer and subjected to a second round of bead enrichment.

After the second bead enrichment, 5′ mono-phosphate RNAs were selected against in two successive steps. First, to selectively degrade RNAs with a 5′ mono-phosphate, NRO RNA was resuspended in 16.5 µl DEPC water, mixed with 0.5 µl SUPERaseIN, 2 µl 10× Terminator reaction buffer A, and 1 µl of Terminator 5′ phosphate-dependent exonuclease (TER51020; Epicentre), and incubated at 30°C for 1 hr. The reaction was extracted and precipitated using the standard method (above), and resuspended in 10 µl DEPC water. Second, 5′ mono-phosphate RNAs were dephosphorylated to prevent their participation in subsequent ligation reactions. For this, the RNA was then mixed with 1 µl SUPERaseIN, 14.5 µl DEPC water, 10× Antarctic Phosphatase buffer, 1.5 µl of Antarctic Phosphatase (M0289S; NEB), and incubated at 37°C for 30 min. The reaction was brought to 200 µl with 10 mM Tris-HCl (pH 7.5), 5 mM EDTA, and heat inactivated at 65°C for 5 min. The reaction was then extracted and precipitated using the standard extraction method and resuspended in 10 µl DEPC water.

The 5′ capped RNAs then were prepared for ligation and the final library preparation steps. The NRO RNAs were then split in half and the experimental sample was treated with tobacco acid pyrophosphatase (TAP): 1 µl SUPERaseIN, 15 µl DEPC water, 3 µl 10× TAP buffer, and 1 µl TAP (T19500; Epicentre), with incubation at 37°C for 1 hr. The control reaction was treated identically except for the addition of TAP. The reaction was brought to 200 µl and then extracted and precipitated using the standard method. The TruSeq RNA 5′ Adapter (RA5) (Illumina part # 15013205) was ligated to the 5′ end of the NRO RNA. First, 50 pmol of the 5′ adapter were mixed with NRO RNA, and 2 µl 50% PEG 8000, brought to 14 µl with DEPC water, incubated at 70°C for 3 min and put on ice for 2 min. Then, 2 µl of 10× T4 RNA Ligase I buffer, 2 µl 10 mM ATP, 1 µl SUPERaseIN, and 1.5 µl T4 RNA Ligase I (M0204; NEB) were added and reaction incubated at 22°C for 4–6 hr. The reaction was then brought to 100 µl with binding buffer and subjected to a third round of bead enrichment. After the third enrichment, samples were reverse transcribed, amplified and PAGE purified as described (*Core et al., 2008*), and quantified before submission for sequencing.

## Mapping of GRO-seq and GRO-cap reads

Libraries were sequenced with Illumina's HiSeq 2000 platform. Reads were required to have passed the CASAVA 1.8 quality filtering to be considered further. To remove reads containing the RT-PCR adapter, we used cutadapt version 0.9.5 (http://code.google.com/p/cutadapt/) with the following command (cutadapt -a TGGAATTCTCGGGTGCCAAGG -a AAAAAAAAAAAAAAAAAAAA -z -O 15 -e .1 --minimum-length=30). The remaining reads were trimmed to 30 bp in length and aligned uniquely to the *C. elegans* WS230 genome using bowtie's default settings (version 0.12.7), which permit two mismatches in the first 28 bp.

Because the 5′ base in each read most closely identifies the location of transcriptionally engaged RNA polymerase prior to the run-on, we created pile-ups using only the first base of each read. For GRO-seq, the pile-up of reads mapping to both strands was normalized by the number of millions of reads that mapped uniquely to the genome and was multiplied by 1000 to obtain RPKM (reads per kilobase per million). For GRO-cap, reads were normalized by the same method but were not multiplied by 1000, hence RPM (reads per million). As expected, the GRO-cap signal is enriched at TSSs compared to the GRO-seq signal.

## Evaluation of GRO-seq normalization

Further analysis of our GRO-seq data using an alternative normalization strategy confirmed that X-linked genes are increased in expression, and autosomal genes are decreased in expression in the *sdc-2* mutant. This new analysis evaluated the original normalization strategy. Because expression of genes on the X chromosome is elevated in *sdc-2* mutants, it is difficult to determine whether the autosomes are changed in expression in the *sdc-2* mutant. The proportion of autosomal reads relative to total reads per experiment is lower in the *sdc-2* mutant (75.9%) than in the control RNAi (85.7%). To normalize the autosomal expression between the two conditions, we divided the average expression of all genes in the *sdc-2* mutant by the scaling factor of this difference (75.9%/84.715% = 0.885). We then investigated changes in the *sdc-2* mutant compared to the control for our most inclusive gene

set: WB WS230 genes that are >250 bp in length (see *Figure 5—source data 1*). With this alternative normalization, the median *sdc-2*/control gene expression ratios increase for both X (1.67 becomes 1.88) and autosomes (0.81 becomes 0.91). This procedure changed the proportion of X-linked genes that are more highly expressed in the mutant (genes increased in expression by 1.5-fold or greater) from 64.9% to 78.4%. The original normalization showed that fewer autosomal genes increase by 1.5-fold than decrease by 1.5-fold (5.2% compared to 26.7%, respectively), and the alternative normalization shows the same trend (8.4% compared to 16.5%). Therefore, the same conclusions about X and autosomal gene expression can be made with either normalization strategy.

## Creation of the unique mappability track

We computationally identified all 30mers in the WS230 genome. After passing these sequences through the cutadapt parameters outlined above, we aligned the remaining sequences uniquely using bowtie. We mapped the 5′ base to determine whether a read beginning at that base pair can be aligned uniquely.

## Gene lists and DNA coordinate conversion

Genome annotation files were downloaded from WB. Protein coding genes that were labeled as 'Coding_transcript' and 'gene' were extracted from the WS230 annotation file. Genes encoding microRNAs and snoRNAs that were labeled as 'miRNA_primary_transcript' and 'snoRNA_mature_transcript', respectively, were extracted from the WS225 annotation file. tRNAs predicted by 'tRNAscan-SE-1.23' were extracted from the WS230 annotation file. The WB remap and unmap tools were used to convert DNA coordinates between releases, thereby ensuring that all analyses matched the correct genome versions.

## Identification of TSSs

### Protein coding genes

As the TSSs for *C. elegans* trans-spliced genes, which comprise 70% of all genes (*Allen et al., 2011*), were unknown, it was essential to properly annotate TSSs for use in our later analyses. We calculated TSSs in wild-type embryos, starved L1s, L3s, and *sdc-2* mutant embryos independently. To do this, we utilized both our GRO-cap and GRO-seq data with the assumption that a true TSS would be supported by a continuous GRO-seq signal extending upstream from the WB start to the newly annotated TSS. To correct for background noise in the TAP+ GRO-cap signal, we subtracted the TAP− control GRO-cap signal from the TAP+ GRO-cap signal. The Z-score was calculated at each position on a chromosome using the standard deviation and mean calculated for that individual chromosome, ignoring the top 0.005% of scores as outliers. Removing these outliers was required so that the locations with the strongest GRO-seq signal did not markedly affect the standard deviation and Z-score. To prevent biasing TSS calls to only those genes in which transcription began at only one base pair, we averaged the Z-score data in a 10 bp moving window with a 1 bp step.

Because the true location of most TSSs are unknown, we searched for TSSs of protein coding genes in a window that began 250 bp downstream of the WB WS230 start and continued upstream until the beginning of the closest transcript (protein coding, microRNA, snoRNA, snRNA, tRNA, or rRNA) on the opposite strand or until 100 bp away from the end of the closest gene on the same strand. If the TSS of a gene was contained within the transcript of another gene or had another protein coding gene within it on the same strand, we did not consider it. This reduced our possible gene list from 20,516 genes to 20,008 genes. Within these genes, we searched for the highest 10 bp Z-score average in the upstream window and concurrently calculated GRO-seq signal across a 200 bp sliding window with a 1 bp step. If the average GRO-seq signal dropped below 1 RPKM, we did not proceed further upstream to identify a TSS. This step was necessary to eliminate TSS calls that were clearly not supported by GRO-seq data. After identifying the highest 10 bp Z-score average, we mapped the highest 1 bp Z-score within the 10 bp window to be called as the TSS position.

To determine both the correct 10 bp Z-score average to use as a cutoff for calling TSSs and the effectiveness of the script at calling TSSs, we visually inspected all TSS calls across the X chromosome and determined whether the GRO-cap and GRO-seq signal supported the call. For each stage, we selected a Z-score cutoff for which >90% of TSS calls were verified by visual inspection (wild-type embryo >4.5, *sdc-2* mutant embryo >4.725, wild-type starved L1 >4.764, wild-type L3 >5.01). To confirm these cutoffs held true across the genome, a random set of 250 autosomal genes with a 10 bp Z-score average greater than the 90th percentile was investigated visually for the wild-type embryo

data set. These genes had >90% agreement for TSS calls, confirming that the same cutoffs could be used for genes on both the X and autosomes. For TSS calls that differed by more than 2 kb upstream from the WB start, we inspected each gene visually to eliminate any TSS calls due to such complications as unannotated transcripts. In addition, for all genes that had an overlapping gene on the opposite strand, we verified each TSS visually. If no TSS had been called, but one could be identified visually with ease, we annotated the TSS. In a few cases, visual inspection allowed us to find a TSS significantly downstream of the WB-annotated start. The criteria used to accept the downstream TSS included a strong GRO-cap signal, no GRO-seq signal upstream of the GRO-cap signal, and no contradicting GRO-cap signal in any worm stage.

In total, we identified a TSS in at least one developmental stage for 31.7% (6353 genes) of all *C. elegans* protein coding genes. Not all TSS calls could be made from each stage. We called TSSs for 4246 genes in wild-type embryos, 2443 genes in wild-type starved L1, 4513 genes in wild-type L3, and 2809 genes in DCC mutant embryos. Of these calls, 875 were unique to wild-type embryos, 290 to wild-type starved L1, 1039 to wild-type L3, and 149 to DCC mutant embryos.

For many analyses, GRO-seq signal and GRO-cap signal were only analyzed across genes for which a TSS had been identified in that stage (*Figure 1F,G* and *Figure 1—figure supplements 5–7*, and *Figure 4—source data 1*). For other analyses (*Figure 6B*), the genes had to have a TSS call from each of the three wild-type stages, and the distances between TSSs among the different stages could not exceed 100 bp. The TSS position was used from the stage that had the highest 10 bp Z-score. For dosage compensation comparisons, a total of 4547 TSS calls was used (4246 TSS calls from wild-type embryos and the 301 TSS calls from *sdc-2* mutant embryos not in the wild-type embryos).

## Non-coding RNAs

Upon obtaining GRO-seq and GRO-cap data, we found that the WB gene models (WS225 models with their coordinates converted to WS230) for many small RNAs, including microRNAs and snoRNAs, lack properly annotated TSSs. As the regulation of expression of many ncRNAs, particularly microRNAs, is essential for proper development, we set out to properly annotate TSSs for many small ncRNAs. We utilized the protein coding gene strategy outlined above to identify likely TSS positions for these ncRNAs. As relatively few of these small ncRNAs exist in the genome, we examined the predicted TSS of each ncRNA gene visually to confirm or deny the computational TSS or to call a TSS. Through these analyses, we annotated TSSs for 74/141 snoRNAs and 70/180 microRNAs, including the five polycistronic microRNA clusters (*mir-35-41*; *mir-42-44*; *mir-229* and *mir-64-66*; *mir-54-56* [*Figure 1B*]; and *mir-73-74*) (*Lau et al., 2001*). We found that most microRNAs are dosage compensated. The compensated microRNAs are the following: *mir-47*, *mir-49*, *mir-54-56* (polycistronic), *mir-62*, *mir-63*, *mir-73-74* (polycistronic), *mir-75*, *mir-81*, *mir-82*, *mir-239.1*, *mir-239.2*, and *mir-791*.

## GRO-seq gene body, and 5′ pausing and 3′ pausing calculations

### GRO-seq gene expression for genes >1.1 kb

The gene expression level for a particular gene was calculated from the average GRO-seq signal within the body of the gene. To prevent the complication of factoring any 5′ or 3′ pausing into our gene expression calculations, we excluded the first and last 300 bp of the gene from the calculations. Within the remaining region, we totaled the GRO-seq signal at each base pair that could be mapped uniquely, and divided this total signal by the number of uniquely mappable bases within the region. To ensure that we averaged GRO-seq signal over a sufficiently large number of bases, we required that genes be ≥1.1 kb in length and have at least 250 uniquely mappable bases within the gene body region included in the calculation. To compare expression from *sdc-2* mutant and control RNAi embryos, we calculated differential gene expression using the R package DESeq (*Anders and Huber, 2010*).

### 5′ Pausing ratio

To investigate the level of 5′ pausing, we divided the average GRO-seq signal in the 5′ end by the average GRO-seq signal in the gene body for genes ≥1.1 kb. To determine the average 5′ end GRO-seq signal, we determined the total GRO-seq signal in the region between 50 bp upstream and 100 bp downstream of the newly annotated TSSs, and divided it by the total number of uniquely mappable bases in the 150 bp region. We then divided this average 5′ GRO-seq signal by the average gene body expression level as calculated above. To be considered paused, we required

that genes have an average gene body expression of ≥1 RPKM, and have a 5′ pausing ratio ≥2. We calculated 5′ pausing ratios for genes in every developmental stage using the TSS calls in the respective stage.

As 5′ pausing is rare and the genome is dense, many genes called paused by the above criteria may not actually be paused if the signal is due to transcription from another source such as 3′ accumulation from an upstream gene or antisense transcription from a bidirectional promoter. To eliminate these false or ambiguous pausing calls, we visually inspected each gene with a 5′ pausing ratio ≥2 in any state to determine whether the level of GRO-seq signal at the 5′ end could be due to transcription from another source. Those that were ambiguous or were likely due to unrelated transcription were removed from our analyses. We found the following ratios of genes (paused/total) to exhibit 5′ pausing: 15 of 3975 genes of wild-type embryos, 14 of 3984 genes of RNAi control embryos, 32 of 3969 genes of *sdc-2* mutant embryos, 166 of 2133 genes of starved L1 larvae, and 78 of 3899 genes of L3 larvae.

RNA Pol II ChIP-seq has been used to assess 5′ accumulation of Pol II binding during L1 arrest (*Baugh et al., 2009*). Though 5′ accumulation was reported, pausing of elongation was not demonstrated. There is little overlap between our set of genes paused during L1 arrest and those reported to have 5′ accumulation of Pol II. This discrepancy is due to the facts that the correct TSSs were not then known for many of the genes and that much of the Pol II accumulated at the 5′ end had not begun elongation and was therefore not detected by GRO-seq (R Baugh, personal communication).

## 3′ Pausing ratio

To investigate the level of 3′ accumulation of Pol II, we calculated a 3′ pausing ratio by dividing the average GRO-seq signal in the 3′ end by the average GRO-seq signal in the gene body. The average GRO-seq signal for the 3′ end was calculated for 200 bp sliding windows with a 1 bp step from 250 bp upstream to 750 bp downstream of the WB WS230 stop. Within each window, the GRO-seq signal was summed and divided by the total number of uniquely mappable bases within the window. The highest average GRO-seq signal was divided by the average gene expression as calculated above to create the 3′ pausing ratio. We then plotted a histogram of 3′ pausing ratios for all genes with an average gene body expression of ≥1 RPKM.

## GRO-seq gene expression for genes greater than 250 bp

To expand the set of genes in which we compared gene expression between *sdc-2* mutant and control RNAi embryos, we calculated gene expression across all genes >250 bp (see *Figure 5—source data 1*). For this calculation, we totaled the GRO-seq signal at each base pair that could be mapped uniquely across the entire length of the gene model, and divided this total signal by the number of uniquely mappable bases within the region. To ensure that we averaged the GRO-seq signal over a sufficiently large number of bases, we required that genes have at least 250 uniquely mappable bases within the gene body.

## Non-coding RNA expression

To investigate the effect of a DCC mutation on non-coding RNA expression, we calculated the average GRO-seq signal across microRNAs and tRNAs. For microRNAs, we totaled the GRO-seq signal from the TSS to the end of primary transcript, including all downstream genes in a polycistronic cluster. We then divided this by the total number of uniquely mappable bases within the window. For tRNAs, we totaled the GRO-seq signal from the start to 50 bp downstream of the stop. We then divided the total signal by the total number of uniquely mappable bases within the window. We included the 50 bp downstream of the stop in our calculations for two reasons. The tRNAs are very similar and have a low level of unique mappability. In expressed tRNAs, GRO-seq signal is evident downstream of the end of the mature transcript. We only included tRNAs with at least 25 uniquely mappable base pairs in our expression values.

## Elongation density index

To determine whether dosage compensation specifically affects transcription elongation across the X chromosome, we determined an elongation density index for each gene with a newly annotated TSS. We calculated the average GRO-seq signal across the last 75% of the gene and divided it by the average GRO-seq signal across the first 25% of the gene, excluding the first and last 500 bp of the gene from this calculation. To ensure that we averaged the GRO-seq signal over a sufficiently large

number of bases, we required that genes be ≥2 kb in length. To avoid outlier ratios that can result from a low number of reads, genes with an average RPKM <1 in the first 500 bp, or the first 25% or last 75% of the remaining gene were excluded from the analysis. To reduce the possibility that the elongation density index was influenced by the 3′ accumulation of Pol II of an upstream gene, we required that genes lack another gene on the same strand within 1 kb of the TSS. We analyzed 481 X-linked genes and 1861 autosomal genes.

## Creating metagene profiles

To compare GRO-seq signal across genes, we scaled genes to be the same length, allowing us to average the GRO-seq signal across them. To avoid small genes that could affect the sensitivity of our analyses, we required that genes be ≥1.5 kb in length. These genes were scaled to the same length as follows: the 5′ end (1000 bp upstream to 500 bp downstream of the TSS) and the 3′ end (500 bp upstream to 1000 bp downstream of the WB stop site) were not scaled, and the remainder of the gene was scaled to a length of 2 kb. We predicted that leaving the ends of the gene unscaled might allow us to better identify any effects that occurred at the ends of genes.

## Calculation of average GRO-seq signal

To investigate GRO-seq trends surrounding the TSS or across scaled metagenes, we plotted the average GRO-seq signal across these regions of interest. To do so, we totaled the strand-specific GRO-seq signal for every gene in the gene list at each base pair in the region. We then divided the total GRO-seq signal at each base pair by the number of genes that are uniquely mappable at that base pair. We then took a 25 bp moving window average of this average GRO-seq signal.

## Heat maps

We used the *Python* package matplotlib to produce heat maps showing the GRO-seq expression of the TSSs of all genes, and the difference in expression between DCC mutant and control RNAi embryos. To show the GRO-seq signal at either the empirically determined TSS or WB start, the signal from each gene with a TSS call was plotted, one gene per row, and the GRO-seq signal was averaged across 15 bp windows. Genes were ordered from top to bottom with increasing distance between the WB start and the TSS called from GRO-cap. To show expression changes in the DCC mutant, we scaled each gene ≥1.5 kb to the same length as described above. We then totaled the GRO-seq signal from the DCC mutant embryos and separately from control RNAi embryos across 100 bp windows and calculated an average. The $\log_2$ of the ratio was plotted for every 100 bp bin across the length of the metagene.

## Calculation of upstream divergent transcription

To calculate the relative level of upstream divergent transcription for each gene, we determined the maximum sense and antisense transcription in a 150 bp window within 500 bp of the TSS. Upstream divergent transcript data from Human and *Drosophila* samples were obtained (***Core et al., 2012***). We used R to plot kernel density estimations of the $\log_2$ (sense/antisense) ratio for *C. elegans*, Human, and *Drosophila*. To determine how far upstream the closest upstream divergent gene was to each TSS, we searched for the closest transcript (protein coding, non-coding RNA, tRNA, or rRNA) upstream and antisense to the TSS. Prior to the search, the 6353 protein coding genes with new TSS calls were re-annotated with the most upstream TSS identified. To determine the upstream distance between the TSS and the start of upstream divergent transcription, we searched 500 bp upstream of the TSS to identify the position with the highest level of GRO-cap TAP+ minus the TAP− signal.

## DCC ChIP-seq

Wild-type N2 animals were grown on NG agar plates with HB101 bacteria. Mixed-stage embryos were harvested from gravid hermaphrodites, and cross-linked with 2% formaldehyde for 30 min. Cross-linked embryos were resuspended in 3 ml of FA buffer (150 mM NaCl, 50 mM HEPES-KOH (pH 7.6), 1 mM EDTA, 1% Triton X-100, 0.1% sodium deoxycholate, 1 mM DTT, protease inhibitor cocktail) for every 1 g of embryos. This mixture was frozen on liquid nitrogen, then ground under liquid nitrogen by mortar and pestle. Chromatin was sheared by the Covaris S2 sonicator (20% duty factor, power level 8, 200 cycles per burst) for a total of 30 min processing time (60 s ON, 45 s OFF, 30 cycles).

To perform the ChIP reactions, extract containing approximately 2 mg of protein was incubated in a microfuge tube with 6.6 µg of anti-DPY-27 or random IgG antibodies overnight at 4°C. A 25 µl

bed volume of protein A Sepharose beads was added to the ChIP for 2 hr. ChIPs were washed for 5 min at room temperature twice with FA buffer (150 mM NaCl), once with FA buffer (1 M NaCl), once with FA buffer (500 mM NaCl), once with TEL buffer (10 mM Tris-HCl (pH 8.0), 250 mM LiCl, 1% NP-40, 1% sodium deoxycholate, 1 mM EDTA), and twice with TE buffer. Protein and DNA were eluted twice with 1% SDS, 250 mM NaCl, 1 mM EDTA at 65°C for 15 min. After elution, sequencing libraries were prepared as published (*Zhong et al., 2010*) with minor changes: sequencing adapters were as described (*Lefrancois et al., 2009*) and adapters were ligated using the NEB Quick Ligation Kit (M2200).

Libraries were sequenced on the Illumina GA2 platform. After barcode removal, 32 bp reads were aligned uniquely to the *C. elegans* WS190 genome using bowtie. MACS (version 1.4) was used to call peaks and create pileups with DPY-27 ChIP as the treatment and random IgG ChIP as the control. To account for read depth, the ChIP signal was normalized to the total number of millions of reads that uniquely aligned to the genome. To correct for non-specific binding, the IgG signal was subtracted from the DPY-27 signal. The resulting ChIP-seq signal from two biological replicates was averaged at each base pair genome-wide.

### Re-analysis of RNA polymerase II ChIP-chip

Raw RNA Pol II ChIP-chip data from experiments using 8WG16 antibody (raised against the hypophosphorylated Pol II C-terminal domain) (*Pferdehirt et al., 2011*) was obtained from GEO (accession numbers GSM634580 and GSM634582). The ChIP signal was normalized to the GC content of individual probes using MA2C (Song et al., 2007). The average ChIP-chip signal surrounding the TSS was calculated using the sitepro script within the CEAS package version 1.0.2 (http://liulab.dfci.harvard.edu/CEAS).

### DNA motif searches

To determine whether *C. elegans* genes contain known core promoter motifs such as the TATA-box and Initiator element (Inr), we performed motif searches using MEME (http://meme.ebi.edu.au/meme/intro.html). To identify a worm TATA-box, we searched for strand-specific motifs within a region 15–45 bp upstream of the TSS. To identify a worm Inr, we searched for strand-specific motifs within 10 bp of the TSS in either direction. These searches identified a TATA consensus of TATAWAWR, compared to TATAWAWR for yeast (*Rhee and Pugh, 2012*), and an Inr consensus of YCAYTY, compared to YYANWYY in humans and TCAYTY in *Drosophila* (*Juven-Gershon and Kadonaga, 2010*). To determine where these motifs lie, we calculated their distance from the TSS. For TATA, we calculated how far upstream the most 5' base lies. For the Inr, we calculated how far the adenine lies from the TSS. In other organisms the adenine has been shown to be the +1 nucleotide in transcription; the location of the worm Inr relative to the TSS suggests that this is true in *C. elegans*.

## Acknowledgements

We thank the Vincent J Coates Genomics Sequencing Laboratory for Illumina sequencing, D Stalford for assistance with figures, and T Cline for comments on the manuscript.

## Additional information

### Funding

| Funder | Grant reference number | Author |
|---|---|---|
| National Institutes of Health | R01GM30702 | Barbara J Meyer |
| Howard Hughes Medical Institute | | Barbara J Meyer |
| National Institutes of Health | T32GM07127 | William S Kruesi |
| National Institutes of Health | R01GM25232 | John T Lis |
| National Institutes of Health | HG004845 | John T Lis |

The funders had no role in study design, data collection and interpretation, or the decision to submit the work for publication.

## Author contributions
WSK, LJC, Conception and design, Acquisition of data, Analysis and interpretation of data, Drafting or revising the article; CTW, Acquisition of data; JTL, BJM, Conception and design, Analysis and interpretation of data, Drafting or revising the article

## Additional files

### Major datasets

The following dataset was generated:

| Author(s) | Year | Dataset title | Dataset ID and/or URL | Database, license, and accessibility information |
|---|---|---|---|---|
| Kruesi WS, Core LJ, Waters CT, Lis JT, Meyer BJ | 2013 | Condensin controls recruitment of RNA polymerase II to achieve X-chromosome dosage compensation | GSE43087; http://www.ncbi.nlm.nih.gov/geo/query/acc.cgi?acc=GSE43087 | In the public domain at GEO: http://www.ncbi.nlm.nih.gov/geo/. |

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
