## [Decision Letter]

Thank you for sending your work entitled “Condensin controls recruitment of RNA Polymerase II to achieve nematode X-chromosome dosage compensation” for consideration at *eLife*. Your article has been favorably evaluated by a Senior editor and 3 reviewers, one of whom is a member of our Board of Reviewing Editors.

The Reviewing editor and the other reviewers discussed their comments before we reached this decision, and the Reviewing editor has assembled the following comments to help you prepare a revised submission.

This study describes the development of a new technique for mapping nascent TSS;GRO-cap. This powerful new technique has been applied to two aspects of *C. elegans* transcriptional regulation: mapping the nascent TSS of outrons and transcriptional regulation of X-chromosome inactivation. Both of these applications are important and publishable. However the first application needs further work/clarification.

1) Our main concern is the authors’ claim that a combination of GRO-seq and GRO-cap allows unequivocal assignment of new upstream TSS for *C. elegans* outrons (5’ end of mature mRNA is formed by SL1 trans-splicing and so doesn’t define gene TSS). For example, in the Discussion section, the authors state that they assign upstream TSS “by requiring that TSS calls be supported by uninterrupted GRO-seq signal for transcriptionally engaged Pol II between the GRO-cap TSS and the previously annotated 5’ end”. GRO-seq shows the presence of active polymerases, and hence, uninterrupted signal does not rule out the possibility of signal arising from multiple, partially overlapping transcripts. For example, if two tandem transcriptional units were located in close proximity, GRO-seq signal could appear continuous (especially as run-on can proceed after the TTS). This could explain the differences with the [10] data, which proposes that separate transcription units may exist upstream of outrons that could be enhancer derived transcripts (eRNAs).

Since this is a key issue we recommend further analysis on this point. To distinguish between these two possibilities, it would be necessary to analyse secondary promoters detected by GRO-cap and confirm that there are no 3’ ends at these positions (PMID: 20522740 or 21085120) that would break these very long outrons into two independent units.

Specific comments related to point 1: A) In Figure 1, one can observe a secondary TSS in the position of the annotated WB start, in addition to the one described by the authors. Specifically, in the right panel (where the reads are centered on the GRO-cap determined TSS), a clear white line moving towards the right is apparent. To be able to appreciate how important this secondary TSS (located in the WB defined position) is with respect to the new ones defined by the authors, it would be useful make the same plot but using the GRO-cap data.

B) Figure 1—figure supplement 11: the fact that GRO-seq signal increases as it passes GRO-cap spikes does not prove that they are continuous transcripts with different 5’ UTRs. Partially overlapping tandem eRNA with lower expression level than the main transcript could produce the same pattern.

C) It seems to us that there is one limitation for the GRO-cap method that the authors did not discuss. Only elongating polymerases in the proximity of the TSS (e.g., <500bp?) will have a nascent RNA short enough to produce a sequencing library compatible with Illumina technology. Although that does not alter the discussed results in this case, this limitation should be stated for future users of the technique.

2) A way to further strengthen the argument that authentic TSS of outrons in many cases is distant to the mature mRNA 5’ end would be ChIP based analysis using Pol II CTD ser5 and ser2 specific antibodies. Thus could be performed on a few selected transcription units with amplicons covering the outron and 5’ half of the coding region.

3) Regarding the accumulation of Pol II at the 3’ end of genes. The authors suggest that extensive pausing at the 3’ end of genes may be linked to trans-splicing. In this context, it would be beneficial if the “not in operon genes” in the analysis presented in Figure 4 were further subdivided into monocistronic trans-spliced genes and non-trans-sliced genes. It would be interesting to see if non-trans-spliced genes also show Pol II accumulation at the 3’ end. In addition they should comment on the possibility that the high U content in the intergenic regions within operons (ur element that direct trans-splicing) and perhaps also at the 3’ end of genes could create a partial bias during GRO-seq and so skew the results for these regions.

4) The effect of *scd2* mutant on dosage compensation and consequent effects on transcription uncovers that dosage compensation also affects small non-coding RNAs (miRNAs). Are there more miRNA affected by dosage compensation? What do they have in common? Do they regulate a group of genes with similar function?

---

## [Author Response]

*1) Our main concern is the authors’ claim that a combination of GRO-seq and GRO-cap allows unequivocal assignment of new upstream TSS for C. elegans outrons (5’ end of mature mRNA is formed by SL1 trans-splicing and so doesn’t define gene TSS). For example, in the Discussion section, the authors state that they assign upstream TSS “by requiring that TSS calls be supported by uninterrupted GRO-seq signal for transcriptionally engaged Pol II between the GRO-cap TSS and the previously annotated 5’ end”. GRO-seq shows the presence of active polymerases, and hence, uninterrupted signal does not rule out the possibility of signal arising from multiple, partially overlapping transcripts. For example, if two tandem transcriptional units were located in close proximity, GRO-seq signal could appear continuous (especially as run-on can proceed after the TTS). This could explain the differences with the [10] data, which proposes that separate transcription units may exist upstream of outrons that could be enhancer derived transcripts (eRNAs)*.

*Since this is a key issue we recommend further analysis on this point. To distinguish between these two possibilities, it would be necessary to analyse secondary promoters detected by GRO-cap and confirm that there are no 3’ ends at these positions (PMID: 20522740 or 21085120) that would break these very long outrons into two independent units*.

*Specific comments related to point 1: A) In Figure 1, one can observe a secondary TSS in the position of the annotated WB start, in addition to the one described by the authors. Specifically, in the right panel (where the reads are centered on the GRO-cap determined TSS), a clear white line moving towards the right is apparent. To be able to appreciate how important this secondary TSS (located in the WB defined position) is with respect to the new ones defined by the authors, it would be useful make the same plot but using the GRO-cap data*.

*B) Figure 1—figure supplement 11: the fact that GRO-seq signal increases as it passes GRO-cap spikes does not prove that they are continuous transcripts with different 5’ UTRs. Partially overlapping tandem eRNA with lower expression level than the main transcript could produce the same pattern*.

*C) It seems to us that there is one limitation for the GRO-cap method that the authors did not discuss. Only elongating polymerases in the proximity of the TSS (e.g., <500bp?) will have a nascent RNA short enough to produce a sequencing library compatible with Illumina technology. Although that does not alter the discussed results in this case, this limitation should be stated for future users of the technique*.

The main concern of the reviewers was our claim that the combination of GRO-cap and GRO-seq permits accurate assignment of new upstream TSSs for *C. elegans* outrons. The reviewers were specifically concerned with our procedure of “requiring that TSS calls be supported by uninterrupted GRO-seq signal for engaged Pol II between the GRO-cap TSS and the previously annotated 5’ end.” They felt that GRO-seq shows the presence of active polymerases and therefore uninterrupted signal does not rule out the possibility of signal arising from multiple, partially overlapping transcripts. The example given was that “if two tandem transcription units were located in close proximity, GRO-seq signal could appear continuous (especially as run-on can proceed after the TTS.” They felt that such overlapping transcripts could explain the difference between our work and that of [10], which proposes that separate transcription units may exist upstream of outrons and could be enhancer derived transcripts. The reviewers asked for further analysis to distinguish between these possibilities. The first request was to analyze the outron regions for 3’ ends as defined in two separate studies, Jan et al., 2011 and [47].

We performed this analysis and found 3’ UTRs and polyA signals to be rare in outrons of greater than 1 kb in length. From 565 such outrons, only 1.4% had an identified 3’ UTR in the Jan et al. (2011) study, and 0.7% had a 3’ UTR in the [47] study. Furthermore, only 3.5% had a polyA site (47). This analysis strongly supports the view that the engaged Pol II signal is not from independent transcription units. Furthermore, in the long outrons, we failed to find the typical GRO-seq signature of *C. elegans* 3' ends: a spike of high Pol II accumulation at the 3’ cleavage and polyadenylation site.

Relevant to this first concern, the reviewers commented that in the heat maps of the old Figure 1 (new Figure 1), they saw a clear white line moving towards the right and interpreted it as secondary TSSs. The reviewers asked that we plot GRO-cap signal relative to our called TSSs to see whether we could find secondary TSSs.

The idea of the plot was helpful and the plot itself was revealing. The heat map of individual genes displaying GRO-cap signal relative to TSSs (new Figure 1) showed that a dominant TSS contributes the majority of the vast GRO-cap signal and thereby supports our claim about the correct identification of TSSs. In addition, we have an alternative interpretation of the light line. It reflects reduced GRO-seq signal caused by trans-splicing near the Wormbase start sites (the sites for the trans-splice acceptor site).

When we first submitted our paper for review, the data from [10] were not available for comparison. For that reason we could only comment on their enhancers for the single gene shown in one of their figures (old Figure 1—figure supplement 11). We found strong overlap between the regions they called enhancers and our TSS calls. That gene was particularly complex for such a comparison because it appears by our analysis to have multiple, alternative TSSs. The interpretation of multiple TSSs rather than independent upstream transcription units has subsequently been supported by finding no evidence of 3’ UTRs or polyA sites from the data of Jan et al. (2011) and [47], and continuous ChIP signal for the hypo-phosphorylated form of Pol II (see new Figure 1—figure supplement 12).

Since the Chen et al. data became available, we found other specific genes to compare. Particularly informative are genes for which Chen et al. call a single upstream enhancer and we called a single strong TSS at exactly the same position as the enhancer. An example is in the new Figure 1—figure supplement 12. It shows a gene with a single called TSS (2534 bp upstream of WB start) that corresponds with the single called enhancer. Continuous ChIP signal of hyper-phosphorylated Pol II was found between the TSS and the WB start, and no 3’ UTRs or polyA signals were found. This and other similar examples strongly support the view that (1) some regions called enhancers by Chen et al. are actual TSSs and (2) GRO-cap signal paired with continuous GRO-seq signal from WB starts defines authentic TSSs. That said, many of the enhancers they mapped do not correspond to outron TSSs, and we are not dismissing the general enhancer mapping performed by Chen et al. A relevant example illustrating this point is in Figure 1—figure supplement 12, which shows a strong TSS overlapping one of the two called enhancers. On balance, though, we do think the enhancer mapping of Chen et al. needs to be corrected in light of our TSS mapping. The revised text now reflects a more balanced assessment of the Chen et al. analysis as it relates to our TSS mapping based on the newly available data.

*2) A way to further strengthen the argument that authentic TSS of outrons in many cases is distant to the mature mRNA 5’ end would be ChIP based analysis using Pol II CTD ser5 and ser2 specific antibodies. Thus could be performed on a few selected transcription units with amplicons covering the outron and 5’ half of the coding region*.

We have addressed this point by reanalyzing ChIP data from our prior studies (53) and those of modENCODE. In general, we found that regions corresponding to long outrons have continuous ChIP-chip signal from antibodies enriched for either the ser2 phosphorylated form of Pol II or the hypo-phosphorylated form of Pol II (see new Figure 1—figure supplement 8 and Figure 3). These results and the restricted run-on length of ∼100 nucleotides that is typical of GRO-seq reactions (16) indicate that GRO-seq signal corresponds to bound Pol II in vivo and is not an artifact of the nuclear run-on reactions in vitro extending beyond the 3’ ends defined in vivo*.* We now emphasize in the protocol (see Figure 2) that GRO-seq run-ons have been tuned to only extend the length of nascent RNAs by 100 bases on average, thus minimizing the concern that independent transcription units have been artifactually linked.

In summary, our paper now provides multiple lines of evidence showing that the GRO-seq signal between newly called TSSs and previously identified TASs represents legitimate outrons rather than independent overlapping upstream transcripts.

A final issue raised by reviewers about our GRO-cap method is the following: ”It seems to us that there is one limitation for the GRO-cap method that the authors did not discuss. Only elongating polymerases in the proximity of the TSS (e.g., <500bp?) will have a nascent RNA short enough to produce a sequencing library compatible with Illumina technology. Although that does not alter the discussed results in this case, this limitation should be stated for future users of the technique.”

We responded to this request by adding the following comment to the legend of Figure 2, which describes the protocol in detail: “We note that transcripts < 500 bp are captured most efficiently on Illumina sequencing platforms.”

*3) Regarding the accumulation of Pol II at the 3’ end of genes. The authors suggest that extensive pausing at the 3’ end of genes may be linked to trans-splicing. In this context, it would be beneficial if the “not in operon genes” in the analysis presented in Figure 4 were further subdivided into monocistronic trans-spliced genes and non-trans-sliced genes. It would be interesting to see if non-trans-spliced genes also show Pol II accumulation at the 3’ end. In addition they should comment on the possibility that the high U content in the intergenic regions within operons (ur element that direct trans-splicing) and perhaps also at the 3’ end of genes could create a partial bias during GRO-seq and so skew the results for these regions*.

The third request made by the reviewers relates to the accumulation of Pol II at the 3' end of genes. In response to the reviewers, we analyzed 3’ Pol II accumulation in monocistronic genes with and without trans-splicing and presented the results in the text and in the new Figure 1—figure supplement 4. We found that first and middle genes in operons had the highest 3’-pausing ratio. Monocistronic genes with trans-splicing had a slightly higher 3’-pausing ratio than terminal genes in operons. Monocistronic genes with trans-splicing had a higher 3’-pausing ratio than monocistronic genes lacking trans-splicing (Mann-Whitney-U p<e-10). The 3’-pausing ratios for genes in all classes were greater than for Drosophila genes. These results further support our proposal that pausing at 3’ ends is linked to trans-splicing.

Regarding the relationship between U content and 3’ end pausing, we found that Pol II accumulation at 3’ ends does not overlap with U-rich regions at 3’ ends. Therefore, the high GRO-seq signal is not due to selective enrichment of U-rich RNAs (Figure 1—figure supplement 2).

*4) The effect of* scd2 *mutant on dosage compensation and consequent effects on transcription uncovers that dosage compensation also affects small non-coding RNAs (miRNAs). Are there more miRNA affected by dosage compensation? What do they have in common? Do they regulate a group of genes with similar function*?

The fourth request from the reviewers concerned the effect of the *sdc-2* dosage compensation mutation on the expression of X-linked microRNAs. We found most embryonically expressed X-linked microRNAs to be dosage compensated. The targets of these microRNAs have not been determined experimentally and the predicted targets do not represent groups of genes with similar functions.